# Robust Noise Attenuation via Adaptive Pooling of Transformer Outputs

**Greyson Brothers** [1]

## Abstract

We investigate the design of pooling methods used to summarize the outputs of transformer embedding models, primarily motivated by reinforcement learning and vision applications. This work considers problems where a subset of the input vectors contains requisite information for a downstream task (signal) while the rest are distractors (noise). By framing pooling as vector quantization with the goal of minimizing signal loss, we demonstrate that the standard methods used to aggregate transformer outputs, AvgPool, MaxPool, and ClsToken, are vulnerable to performance collapse as the signal-to-noise ratio (SNR) of inputs fluctuates. We then show that an attention-based *adaptive pooling* method can approximate the signal-optimal vector quantizer within derived error bounds for any SNR. Our theoretical results are first validated by supervised experiments on a synthetic dataset designed to isolate the SNR problem, then generalized to standard relational reasoning, multi-agent reinforcement learning, and vision benchmarks with noisy observations, where transformers with adaptive pooling display superior robustness across tasks.

## 1. Introduction

Autonomous systems, both artificial and biological, require diverse and redundant sensors to capture a reliable picture of their environment. Limited processing resources and competitive environments pressure biological systems to process their abundant sensory information efficiently. To satisfy this constraint, only the subset of information relevant to the current task is retained while the rest is considered noise and attenuated – the process of selective attention (Broadbent, 1958; Driver, 2001).

[1]Johns Hopkins University Applied Physics Laboratory, Maryland, USA. Correspondence to: Greyson Brothers <greyson.brothers@jhuapl.edu>.

*Proceedings of the 42$^{nd}$ International Conference on Machine Learning*, Vancouver, Canada. PMLR 267, 2025. Copyright 2025 by the author(s).

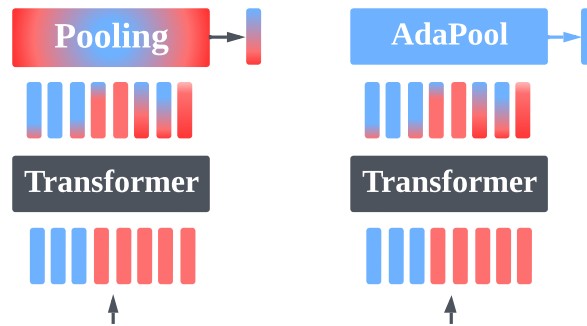

Figure 1: Given a mix of signal (blue) and noise (red) inputs, standard pooling methods result in unwanted interference in the aggregate representation of transformer outputs. AdaPool learns to adaptively attenuate different ratios of noise and obtain a cleaner signal.

Complex reinforcement learning (RL) environments pose similar challenges, especially those seeking to bridge the gap with the real world. Modern approaches have adopted relational architectures, like transformers (Vaswani et al., 2017) and graph neural networks (GNNs) (Battaglia et al., 2018), to process their observations (Zambaldi et al., 2019; Hu et al., 2021; Nayakanti et al., 2023; Huang et al., 2023; Nayak et al., 2023). By breaking each observation into vector-based tokens representing entities, memories, or multi-modal sensory streams, these networks can achieve the following benefits over standard feedforward policy architectures: (1) variable-sized observation spaces enabling scalability (Iqbal & Sha, 2019; Hsu et al., 2021), (2) sample efficiency via permutation invariance and/or equivariance over the tokens (depending on architectural choices) (Liu et al., 2020; McClellan et al., 2024; Hao et al., 2023), and (3) parameter efficiency by learning pairwise relations between tokens explicitly rather than implicitly via redundant feedforward subnetworks (Santoro et al., 2017). In light of the benefits of relational architectures over their feedforward counterparts, we examine a critical implementation detail for transformers that is often overlooked.

Each inference of a standard transformer encoder produces as many output embeddings as input vectors. Under the sequence-to-sequence training paradigm that they were designed for, the next token in the sequence is an obvious target for each output embedding. However, in domains such as computer vision or RL, deciding how to map a set

of output embeddings to a class or action distribution is less straightforward. The goal is to discard irrelevant embeddings while condensing the most salient information from those remaining into a representation that can be used for downstream tasks. Notably, this mirrors the broader aim of pooling in general (Boureau et al., 2010). Indeed, for non-sequential transformer applications, this is usually accomplished via average pooling (Karamcheti et al., 2023; Zhai et al., 2022; Vinyals et al., 2019), max pooling (Zambaldi et al., 2019; Chen et al., 2021), or the use of a learned class token (Devlin et al., 2019; Dosovitskiy et al., 2021; He et al., 2022). However, many prior works treat this step as an arbitrary design choice without theoretical justifications.

In this paper, we derive a framework to evaluate the performance of differentiable vector pooling methods on inputs composed of both "signal" and "noise" vectors. Our analysis reveals that popular approaches like average and max pooling can suffer catastrophic performance drops when their inductive biases are misaligned with the noise level. Motivated by this finding, we explore an attention-based *adaptive pooling* (AdaPool) mechanism that dynamically weights relevant vectors during aggregation, mitigating interference from distractors. We then show that AvgPool, MaxPool, and ClsToken are special cases of AdaPool. Crucially, we prove that AdaPool can approximate the signal-optimal vector quantizer for any signal-to-noise ratio under explicit error bounds derived from the data distribution. We validate these theoretical insights through synthetic supervised experiments and then demonstrate their practical impact on multi-agent reinforcement learning (MPE), relational reasoning (BoxWorld), and vision (CIFAR) benchmarks. Across all tasks, the adaptive pooling method consistently outperforms standard baselines and avoids their failure modes in the presence of significant noise.

The contributions of this work can be summarized as follows:

1. We provide a theoretical framework for analyzing the robustness of vector pooling methods to noisy inputs.

2. We show that attention can robustly pool observations across the full spectrum of noisy inputs, and derive error bounds on its ability to optimally retain signal.

3. We perform extensive experiments on supervised and RL benchmarks to corroborate our theoretical results.

## 2. Related Work

### 2.1. Interference and Associative Memories

The problem of noise robustness posed by this paper is most closely related to the problem of noisy recall in the context of associative memories (AMs). AMs, such as Hopfield

Networks (Hopfield, 1982), are models that store sets of memory vectors. They utilize an update rule analogous to a pooling method that summarizes and returns information from the set of memories according to their relationship with a retrieval cue. Such networks have a storage capacity under which the pooling method is guaranteed to return one of the memory vectors and beyond which it may return a spurious mixture of interfering memories (Pham et al., 2024). This capacity is generally exceeded when the number of memories far exceeds the dimensionality of each vector. A growing line of work on Dense Associative Memories (Krotov & Hopfield, 2016; Demircigil et al., 2017; Hoover et al., 2024) has shown the design of this update/pooling function has significant ramifications on the number of memories that can be pooled without destructive interference. By introducing increasingly sharp non-linear activation functions into the update rule, Dense AMs are able to achieve a memory capacity that is exponential with respect to the dimension of the memories. Ramsauer et al. (2021) showed that an exponential form of this update rule is equivalent to the attention mechanism, going as far as to propose a standalone Hopfield Pooling layer utilizing attention. Taking inspiration from attention-based recall methods, our goal is to extract the most task-relevant information from a set of abundant and diverse sensory embeddings while minimizing destructive interference, as conveyed in Figure 1.

### 2.2. Attention-based Pooling

A number of works have recently utilized cross-attention with a single query vector as a pooling method. Research on memory augmented neural networks yielded the Neural Turing Machine (Graves et al., 2014) and Differentiable Neural Computer (Graves et al., 2016), both using attention to extract information from a set of memory vectors, but without the recall guarantees afforded by Hopfield networks. These methods used the output of a recurrent controller as the query vector. Ilse et al. (2018) use attention-based pooling with a learned query for multi-instance learning. Stergiou & Poppe (2023) apply attention-based pooling to computer vision problems, terming the technique *adaptive pooling* (AdaPool), which we adopt. They use the centroid of the input set as the query for pixel-level pooling. Several works utilize a learned query for vision-based transformers, which is similar to the ClsToken approach but only introduces the learned query in a final aggregation layer (Touvron et al., 2021; Przewięźlikowski et al., 2024; Torres et al., 2024). Finally, Perceiver (Jaegle et al., 2021), Set Transformer (Lee et al., 2019), and Universal Physics Transformers (Alkin et al., 2024) use multiple learned queries as inducing points to reduce the cardinality of an input set for more efficient self-attention blocks. We further discuss the implementation details of adaptive pooling in Section 3.5, and motivate a novel choice of query based on our analysis.

## 3. Methods

In this section, we formally define vector pooling methods and establish a theoretical framework under which they can be evaluated analytically for signal loss under various signal-to-noise regimes. Critically, we demonstrate that attention-based pooling can approximate a signal-optimal vector quantizer for inputs with any noise ratio.

### 3.1. Data and Noise

We define our input domain as a set of vectors containing sensory information. It is often helpful to imagine these vectors as points, particles, or entities. We represent these sets as matrices $\mathbf{X} \in \mathbb{R}^{N \times d}$, where $N$ indicates the cardinality of the set and $d$ reflects the dimensionality of each vector. In a multi-agent RL setting, each vector might contain the x-y positions and velocities of different entities, as well as relevant features including entity type, team, health, etc. In a multi-modal perception example, the set of input vectors may be split between encoded image patches, encoded audio tokens, and embedded text tokens. Abstractly, each vector in the set represents a snapshot from the different sensory streams an agent uses to perceive its environment.

For any given inference, some number $k \leq N$ of those input vectors will contain information that is relevant to the learning task. We label the $k$ task-relevant vectors as the signal subset $\mathbf{X}_s$ and the rest as noise subset $\mathbf{X}_\eta$, such that the input set $\mathbf{X} = \mathbf{X}_s \cup \mathbf{X}_\eta$ and $\mathbf{X}_s \cap \mathbf{X}_\eta = \emptyset$. Under this framework, we define a signal-to-noise ratio $SNR = \frac{k}{N}$.

A vector $\mathbf{x}_i$ belongs to the signal subset when the learning target $y$ is a function of that vector. Formally, $\mathbf{x}_i \in \mathbf{X}_s \iff \frac{\partial y}{\partial \mathbf{x}_i} \neq 0$ and $\mathbf{x}_i \in \mathbf{X}_\eta \iff \frac{\partial y}{\partial \mathbf{x}_i} = 0$. The following sections utilize this notation to lay the groundwork for our theoretical analysis of vector pooling methods.

### 3.2. Vector Quantization

We extend analytical tools developed for vector quantization (VQ) (Gray, 1984) and lossy compression to evaluate the theoretical noise-robustness of pooling methods. It is worth noting that vector quantization should not be confused with the recent weight quantization techniques used to reduce the memory footprint of language models (Jacob et al., 2018). Instead, VQ is concerned with evenly dividing a large set of points into discrete clusters for data compression. The compressed set uses a single vector representation for each cluster in place of the original points. A well-known example is the k-means clustering algorithm (MacQueen, 1967). We frame global vector pooling methods as a degenerate case of vector quantization with a single cluster.

**Definition 3.1.** A *Global Vector Pool* is any differentiable vector quantizer $C(\mathbf{X}) : \mathbb{R}^{N \times d} \to \mathbb{R}^d$ of the following form:

$$C(\mathbf{X}) = \sum_i^N \mathbf{w}_i \odot \mathbf{x}_i \qquad (1)$$

where $\odot$ indicates the Hadamard product with a weight vector $\mathbf{w}_i = [w_{i,1}, ..., w_{i,d}]$ whose elements $w_{i,j} \in \mathbb{R}$. If all elements of $\mathbf{w}_i$ are identical, then the result is equivalent to using a scalar weight $w_i$. We label this function $C$ as it represents a compressor.

In the VQ literature, the information loss incurred by a quantizer is referred to as quantization error or distortion. The standard metric used is mean squared error (MSE) between each point and its cluster's representation. Gray (1984) defines the optimal quantizer as that which minimizes MSE over the input set, yielding the centroid in the case of a single cluster. For our purposes, we care only about signal distortion; noise-related quantization error is of no consequence. Thus, we define a related measure of information loss specifically for our use case:

**Definition 3.2.** The *Signal Loss* $\mathbb{L}$ of a vector pool $C$ on a noisy set $\mathbf{X}$ is the MSE between the compressed representation $C(\mathbf{X}) = \mathbf{x}_c$ and the subset of signal vectors $\mathbf{X}_s \subseteq \mathbf{X}$.

$$\mathbb{L}(\mathbf{X}, \mathbf{x}_c) = \frac{1}{k} \sum_{\mathbf{x}_s \in \mathbf{X}_s} (\mathbf{x}_s - \mathbf{x}_c)^2$$

**Corollary 3.3.** *The point $\mathbf{x}_c^*$ that minimizes signal loss is the centroid of the signal subset (see proof A.1). We say that the global vector pool $C^*$ that computes this point is signal-optimal.*

From Definition 3.2 and Corollary 3.3, it follows that the signal-optimal global vector pool assigns weights

$$w_i = \begin{cases} \frac{1}{k} & \text{if } \mathbf{x}_i \in \mathbf{X}_s \\ 0 & \text{if } \mathbf{x}_i \in \mathbf{X}_\eta \end{cases}$$

### 3.3. Global Average Pooling

**Definition 3.4.** *Global Average Pooling* is a vector quantizer of the following form:

$$AvgPool(\mathbf{X}) = \sum_i^N w_i \cdot \mathbf{x}_i \qquad (2)$$

where weights $w_i$ are are scalars given by

$$w_i = \frac{1}{N} \qquad (3)$$

**Corollary 3.5.** *AvgPool is signal-optimal if the input set contains no noise.*

$$\mathbf{X}_\eta = \emptyset \implies AvgPool = C^*$$

*See proof A.2*

**Corollary 3.6.** *When the input contains noise, $\mathbf{X}_\eta \neq \emptyset$, AvgPool is signal-optimal if and only if the centroid of $X_s$ is equivalent to the centroid of $X_\eta$.*

$$AvgPool = C^* \iff AvgPool(\mathbf{X}_s) = AvgPool(\mathbf{X}_\eta)$$

*See proof A.3*

We note that these two cases are extremely limiting. Average pool is only signal-optimal in the unlikely cases that the entire input set is strictly composed of task-relevant vectors or the signal and noise vectors are identically distributed. When that is not the case, AvgPool tends to yield higher signal loss with each additional noise vector.

### 3.4. Global Max Pooling

**Definition 3.7.** *Global Max Pooling* is a vector quantizer of the following form:

$$MaxPool(\mathbf{X}) = \sum_i^N \mathbf{w}_i \odot \mathbf{x}_i \qquad (4)$$

where the elements of the weight vector $\mathbf{w}_i = [w_{i,1}, ..., w_{i,d}]$ are given by

$$w_{i,d} = \begin{cases} 1 & \text{if } x_{i,d} > x_{j,d}, \forall j \neq i \\ 0 & \text{otherwise} \end{cases} \qquad (5)$$

Any elements whose values are the maximum along a feature column of $\mathbf{X}$ are given a weight of one and all others a weight of zero. For notational simplicity, we make the assumption that a unique maximum exists. If a given column has $m$ elements that share the maximum value, then the MaxPool can still be obtained with this summation by giving each a weight of $\frac{1}{m}$.

**Corollary 3.8.** *MaxPool is signal-optimal if and only if the input set $\mathbf{X}$ contains a single signal vector ($k = 1$) taking the maximal value along each feature dimension.*

$$MaxPool = C^* \iff |\mathbf{X}_s| = 1, MaxPool(\mathbf{X}) = \mathbf{x}_s$$

*See proof A.4*

We observe that MaxPool acts as a complement to AvgPool, where signal loss tends to increase with each additional signal vector added to the input set. They both have inductive biases that result in best performance at opposite ends of the signal-to-noise spectrum and experience increasing signal loss when the SNR changes.

### 3.5. Global Adaptive Pooling

Adaptive pooling utilizes cross-attention with a single query to pool a set of vectors. Prior works have utilized kernel functions and inner product spaces to study the set of relational functions that attention can approximate (Tsai et al., 2019; Altabaa & Lafferty, 2024). Following this line of work, we let $r(\mathbf{x}_q, \mathbf{x}_i) : \mathbb{R}^d \times \mathbb{R}^d \to \mathbb{R}$ be a parametrizable, asymmetric kernel used to compute the relation between a query and a sensory vector. Given parametrizable weight matrices $W_Q, W_K \in \mathbb{R}^{d \times d}$ and a query vector $\mathbf{x}_q \in \mathbb{R}^d$, we define the relation kernel as the scaled dot product

$$r(\mathbf{x}_q, \mathbf{x}_i) = \langle \phi_\theta(\mathbf{x}_q), \psi_\theta(\mathbf{x}_i) \rangle = \mathbf{x}_q W_Q W_K^\top \mathbf{x}_i^\top \cdot \frac{1}{\sqrt{d}}$$

For our purposes, the query $\mathbf{x}_q$ is fixed for all $\mathbf{x}_i$, so we use the shorthand $r(\mathbf{x}_q, \mathbf{x}_i) = r_i$ for notational convenience.

**Definition 3.9.** Given parametrizable weight matrices $W_Q, W_K, W_V \in \mathbb{R}^{d \times d}$ and a query vector $\mathbf{x}_q \in \mathbb{R}^d$, *Global Adaptive Pooling* is a vector quantizer of the following form:

$$AdaPool(\mathbf{X}) = \sum_i^N w_i \cdot \mathbf{x}_i W_V \qquad (6)$$

where each weight $w_i$ is given by the softmax of relation $r_i$

$$w_i = \frac{\exp(r_i)}{\sum_j^N \exp(r_j)} \qquad (7)$$

The choice of the query is thus an inductive bias that controls the nature of the relation kernel, which in turn affects the attenuation of the input set.

**Corollary 3.10.** *AvgPool is a special case of AdaPool.*
*See proof A.6*

**Corollary 3.11.** *MaxPool is a special case of AdaPool.*
*See proof A.7*

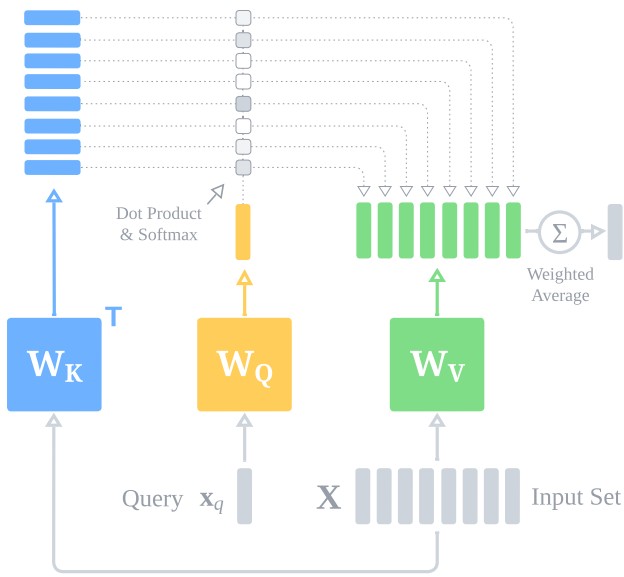

Figure 2: Attention as a vector pooling method (AdaPool).

Next, we construct a set of error bounds on AdaPool's approximation of the signal-optimal quantizer. The general intuition is that the output of the softmax should equally distribute weights amongst the signal vectors and give zero weight to the noise vectors, following from Corollary 3.3. Since softmax normalizes inputs by the sum of their exponentiated values, there needs to be a meaningful margin between signal and noise relation scores to push the noise weights to zero, and the signal scores need to be similar in magnitude such that one does not dominate the others. To formalize this intuition, we introduce the following notation to describe the neighborhood widths $\epsilon$ for signal and noise:

$$\epsilon_s = max\{r_s\} - min\{r_s\} \geq 0$$
$$\epsilon_\eta = max\{r_\eta\} - min\{r_\eta\} \geq 0 \tag{8}$$

We additionally define the minimum margin $M$ and max distance $D$ between signal and noise neighborhoods as

$$M = min\{r_s\} - max\{r_\eta\}$$
$$D = max\{r_s\} - min\{r_\eta\} = M + \epsilon_s + \epsilon_\eta \tag{9}$$

These values are illustrated in Figure 3 using an example set of relation scores for better intuition. These are used to bound the worst-case weights for any given set of inputs.

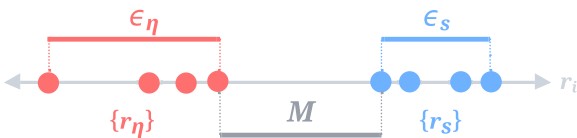

Figure 3: An example displaying the $\epsilon$ neighborhoods of signal (blue) and noise (red) relation values, along with the margin $M$ between the two neighborhoods $\{r_s\}$ and $\{r_\eta\}$. The axis represents the codomain of the relation kernel.

**Theorem 3.12.** *For any signal-to-noise ratio, AdaPool can approximate the signal-optimal vector pool within an error bound determined by the distribution of signal and noise relations. Explicitly, **for any signal vector** $\mathbf{x}_i \in \mathbf{X}_s$, AdaPool approximates the optimal weight of $w_i^* = \frac{1}{k}$ within error bounds $L_s \leq w_i^* - w_i \leq U_s$, which are the following lower and upper bounds respectively:*

$$L_s = \frac{1}{k} - \left(1 + (k-1) \cdot e^{-\epsilon_s} + (N-k) \cdot e^{-D}\right)^{-1}$$

$$U_s = \frac{1}{k} - \left(1 + (k-1) \cdot e^{\epsilon_s} + (N-k) \cdot e^{-M}\right)^{-1}$$

*and **for any noise vector** $\mathbf{x}_i \in \mathbf{X}_\eta$, AdaPool approximates the optimal weight of $w_i^* = 0$ within the error bounds $L_\eta \leq w_i^* - w_i \leq U_\eta$, which are the following lower*

*and upper bounds respectively:*

$$L_\eta = -\left(k \cdot e^M + 1 + (N-k-1) \cdot e^{-\epsilon_\eta}\right)^{-1}$$

$$U_\eta = -\left(k \cdot e^D + 1 + (N-k-1) \cdot e^{\epsilon_\eta}\right)^{-1}$$

*See proof A.5*

*Remark* 3.13. Crucially, as the margin $M$ increases and the signal and noise neighborhoods $\epsilon_s$ and $\epsilon_\eta$ shrink, the bounds squeeze the approximation error to zero. This is influenced by the data distribution, the expressiveness of the relation kernel, and any preprocessing of the input set, such as embedding via transformer.

To make this likely in practice, we propose selecting the query from the signal subset $\mathbf{x}_q \in \mathbf{X}_s$. Since the dot-product relation kernel computes a notion of similarity, when the query is a signal vector, the dot product with other signal vectors should tend to be higher than the dot product with noise vectors. We argue that using the centroid of the whole set as a query (Stergiou & Poppe, 2023) is not robust, as it is influenced by the ratio of signal and noise vectors in a given input. We also argue that selecting a learned embedding (Lee et al., 2019) is not ideal, as it is fixed regardless of changes to the distributions of signal and noise per sample. For example, a simple rotation of all signal and noise points about the origin could severely alter their dot products with a fixed learned embedding, while dot products with a signal vector would be invariant to such transformations.

While choosing a known signal vector for every input might seem like a nebulous task, we provide numerous examples here to ground and inspire such choices. For entity-based RL problems, the network is controlling one of the N entities in its observation space. We use the transformer embedding of that entity's state token as the query since it always contains task-relevant information. If the inputs are memory vectors, the current state of the environment acts as a signal-rich query vector. For vision tasks with structured images, like video games, one might choose a particular image patch to be the query, such as a patch covering a mini-map or status indicators. For real-world vision tasks, a patch from the center of the image may be desirable, as it will generally contain focal content coinciding with the gaze of the agent. We explore the consequences of such choices in Section 4.5.

By using a query from the input set, we also have a natural way of preserving the residual stream by adding the output of AdaPool back to the query. This helps with gradient propagation, and we observe better empirical performance than without extending a skip connection to the pooling layer. Finally, for an analysis of time complexity, see B.1.

### 3.6. ClsToken

Class tokens are a common alternative to the above pooling methods, introduced by BERT (2019) and frequently used in vision applications (Dosovitskiy et al., 2021; He et al., 2022). For this method, a learned parameter vector is appended to the input set before being fed into the transformer encoder. The corresponding embedding is taken from the output and used for downstream tasks. If one ignores the final feedforward sublayer and discards the $N - 1$ other output embeddings produced by the transformer, the ClsToken and AdaPool methods differ only by the choice of query vector $\mathbf{x}_q$. ClsToken takes $\mathbf{x}_q$ to be the output embedding of the learned parameter vector, whereas we construct AdaPool to select the output embedding of a signal vector.

## 4. Experiments

We designed a variety of experiments to evaluate our theoretical findings. Using a large synthetic dataset, we first construct a supervised learning task to isolate the robustness of transformers with each pooling method across the full spectrum of noise levels. The same architectures are then applied to RL tasks in the Multi-Particle Environment and BoxWorld, first mirroring the supervised task and then looking at performance in standard benchmark scenarios with increasing noise. Finally, we test the generalization of our analysis to real-world data with the CIFAR image classification dataset. Critically, we show that AdaPool displays superior robustness across noise levels on all baselines, validating our theoretical results. Code is publicly available at https://github.com/agbrothers/pooling.

### 4.1. Synthetic Dataset

For our first experiments, we generated a synthetic dataset with 1 million samples. Each sample is a set containing $N = 128$ vectors with $d = 16$ features per vector, represented by a 2D array. Each feature column is drawn from a unique distribution, evenly split between randomly parameterized exponential, gaussian, and uniform distributions. The ordering of distributions is shuffled for each sample to ensure diversity and prevent overfitting.

### 4.2. Noise Robustness Experiments

To assess our theoretical findings, we design a supervised experiment to mirror our analytical framework. Given a set of $N$ input vectors representing points in space, the network must predict the centroid of the $k$-Nearest Neighbors (the signal subset) to an arbitrarily chosen target point. The target point is indicated to the network by adding a learned embedding to it at inference time, similar to the use of learned positional embeddings used in language modeling tasks (Radford et al., 2018). As we vary $1 \le k \le N$, we

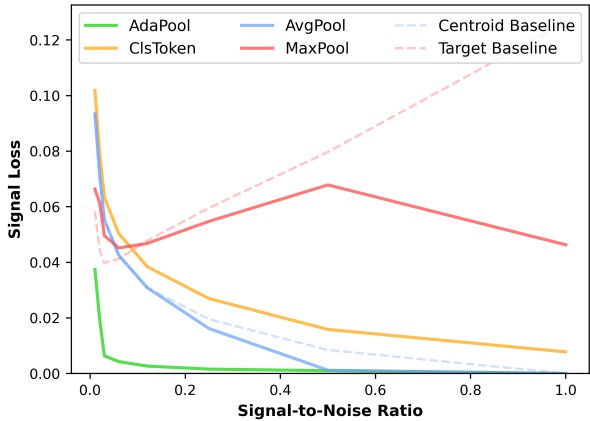

Figure 4: Signal Loss (MSE) on the KNN-Centroid Task (N=128, d=16). Lower is better, data is provided in Table 3.

can explicitly control the signal-to-noise ratio of each observation, allowing us to gauge robustness by measuring signal loss under each noise regime. As baselines, we display the signal loss incurred by naively predicting the centroid of the entire input set, as well as naively predicting the target vector used to source the nearest neighbor subset.

For these experiments, we use 5-fold cross-validation to train a 12-layer transformer encoder capped by a pooling method. All models were on the order of 600k parameters with the same initial weight configuration for the base transformer. Additional hyperparameters are listed in the appendix C.2. We report the signal loss on a test set comprised of 100k holdout samples in Figure 4, averaged across the model checkpoints with the lowest validation loss on each fold. Additional ablations are reported in B.5, varying the dimensionality and cardinality of the inputs as well as the dimensionality and depth of the networks.

AdaPool exhibits the lowest signal loss and most consistent performance across noise regimes. In particular, it exceeds all other approaches by an order of magnitude in the low SNR regimes (0.03-0.25). Predictably, MaxPool has its best relative performance in the lowest SNR regime and rapidly declines as signal increases. Conversely, AvgPool performs best for high SNR (0.5-1.0) and deteriorates predictably as signal becomes sparse. AvgPool also closely follows the naive centroid baseline, while MaxPool closely follows the naive target baseline until an SNR of 1.0, at which point it improves slightly. The ClsToken method performs slightly worse than AvgPool at each noise level. All methods exhibit their worst absolute performance on the highest noise KNN-1 task, with the exception of MaxPool. These findings directly support our theoretical analysis.

### 4.3. Multi-Agent Experiments

To generalize our findings to reinforcement learning, we use the standard Multi-Particle Environment (MPE) benchmark.

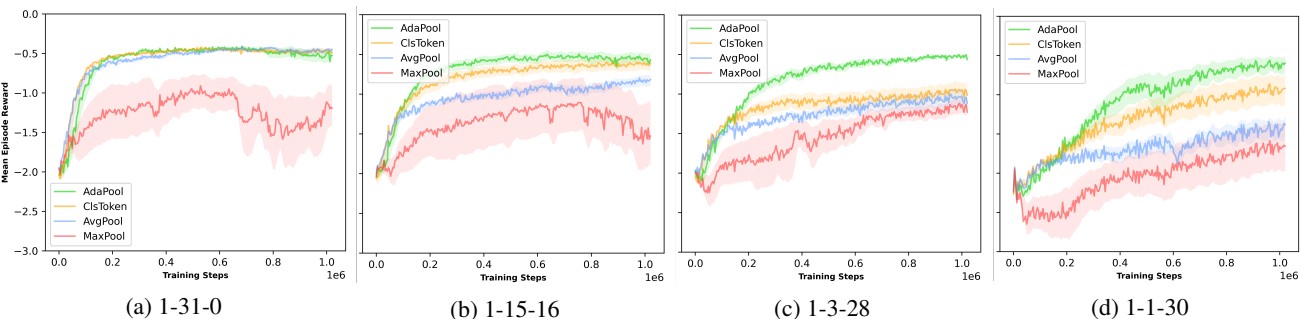

(a) 1-31-0  (b) 1-15-16  (c) 1-3-28  (d) 1-1-30

Figure 5: Performance on the simple centroid environment, with the number of particles labeled Agents-Signal-Noise. The SNR decreases from $\frac{32}{32}$ (a) $\rightarrow \frac{16}{32}$ (b) $\rightarrow \frac{4}{32}$ (c) $\rightarrow \frac{2}{32}$ (d). Lines represent the mean value across seeds, with standard error shaded and the Y-axis fixed across plots to show performance decay.

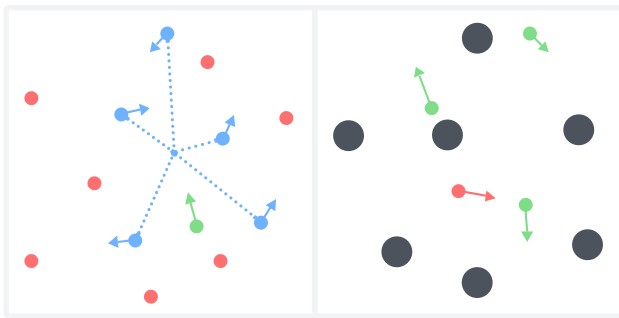

Figure 6: Left, the simple centroid environment with an agent (green), signal entities (blue), and noise entities (red). Right, the simple tag environment with a predator (red), prey (green), and obstacles (black).

The environment provides a simple 2D world with baseline scenarios and an API to implement custom scenarios. We utilize the baseline simple tag environment and implement a custom simple centroid environment, both described below. While the default environment returns a flat observation vector, we process the observation into entity state tokens, such that each entity is represented by an 8-dimensional state vector containing (x,y) position, (x,y) velocity, and a 4-dimensional learned embedding corresponding to the entity type: [self, predator, prey, obstacle]. These tokens are each passed through the same projection layer to map them to the hidden dimension of the network. Expanding the size of the hidden dimension relative to the inputs alleviates the burden on the pooling method to effectively compress information, as found in our ablation studies in B.5. For these experiments, we thus limit the hidden dimension to at most the size of the input tokens.

We use the Proximal Policy Optimization (PPO) (Schulman et al., 2017) reinforcement learning algorithm in an online, centralized-training decentralized-execution (CTDE) setup: each agent is controlled by a copy of the same policy network but observes and acts independently. The training batch compiles experience from all learning agents to update the weights of the single policy, which is then copied

back to each agent. We use 10 seeds per method for each simple centroid experiment and 20 per method for simple tag. The policy architecture mirrors the architecture used in the previous experiments (4.2) with the addition of a pair of 3-layer MLPs that map the pooled output to action logits and value predictions respectively. For each trial, all four methods are run with the same seeds, base transformer weights, and environmental initial conditions. Training batches consist of 8192 samples drawn from 128-timestep episodes, with all experiments training for 4 million timesteps total. Additional training hyperparameters can be found in C.4.

### 4.3.1. SIMPLE CENTROID

We first design a custom MPE scenario to mirror the supervised KNN-Centroid task in a multi-agent RL setting. We sample a set of signal agents that move according to a random policy, along with a set of noise agents whose positions are fixed, as shown on the left of Figure 6. We train a single additional agent with a reward to minimize its distance to the centroid of the signal agents. We use a continuous action space, where the actions are accelerations in the cardinal directions. Results are displayed in Figure 5.

AdaPool consistently reaches the same level of performance across noise levels, while all other methods deteriorate as noise increases. MaxPool was the worst performer by a wide margin across all levels, even when signal was sparse. As in the synthetic experiment, we observe that AdaPool performs the best relative to other methods in the mid-low SNR regime ($\frac{4}{32}$). With all methods, we observe an intuitive correlation between sample efficiency and noise level; more noise requires more samples to reach the same performance.

### 4.3.2. SIMPLE TAG + NOISE

For this experiment, we examine the impact of increasing the volume and ratio of noise in the default simple tag benchmark provided by MPE. Simple tag is a predator-prey scenario, where predators receive +10 reward for colliding with prey and prey receive -10 reward for colliding with

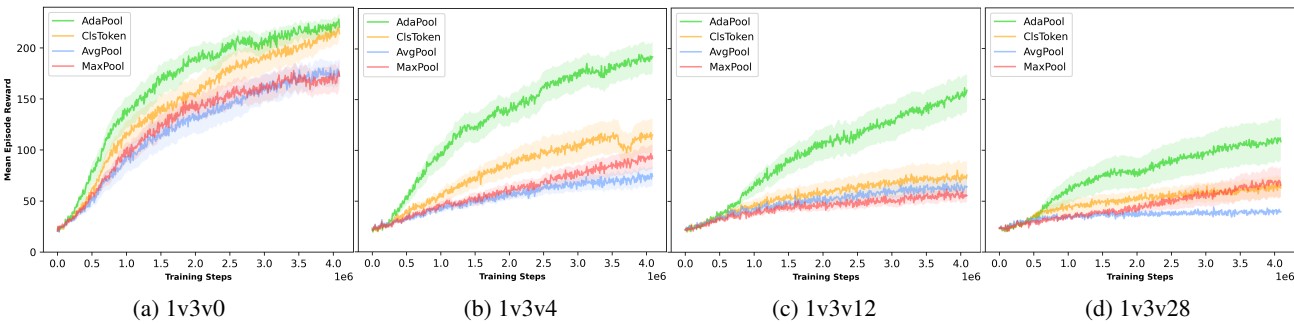

| (a) 1v3v0 | (b) 1v3v4 | (c) 1v3v12 | (d) 1v3v28 |

Figure 7: Scaling the number of distractor obstacles (noise) while keeping reward dynamics fixed. Subplots are labeled Predators-Prey-Obstacles, and the Y-axis is shared across plots.

predators, and collidable obstacles are scattered about, as shown on the right of Figure 6. For this experiment, we turn off obstacle collisions such that they have no influence on the reward function or dynamics of the game. We use a single predator and 3 prey agents, and train with increasing numbers of obstacles acting as noise in the observation space. To enable objective evaluation and prevent competition from introducing non-stationarity into the problem, we only train the predator and control the prey agents with a heuristic that randomly samples a direction to move in. Unlike the simple centroid experiment, we use a discrete action space here.

As shown in Figure 7, the final mean reward of all methods declines significantly as both the volume and ratio of noise increase, indicating a major decrease in sample efficiency within the fixed training budget of 4 million timesteps. Since the reward dynamics are identical across all scenarios and the training time is fixed, this performance decrease can be directly linked to the difficulty of learning from observations with sparse signal. Predictably, AvgPool suffers the worst from an increase in noise, with the final mean reward dropping 77.4%. ClsToken drops a similar 70.4%, MaxPool falls 60.7%, and AdaPool drops 50.9%. AdaPool attains the highest mean reward and lowest performance decline across all noise levels.

### 4.4. BoxWorld

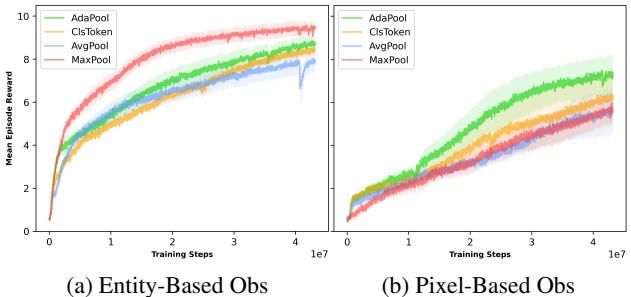

| (a) Entity-Based Obs | (b) Pixel-Based Obs |

Figure 8: BoxWorld performance on entities only (8) vs all pixels (50). The Y-axis is shared across plots.

BoxWorld is a vision-based relational reasoning task introduced by Zambaldi et al. (2019). It involves picking up keys to unlock a sequence of boxes in the correct order to reach a goal gem. There are also distractor paths that lead to dead ends, requiring the agent to reason over which key to pick up or which box to unlock next. Keys, Locks, and Boxes are represented by pixels on a 2D grid, as depicted in Figure 10. We train on an instance with a goal sequence of length 2, 2 distractors, and a 7x7 grid under two observation regimes. One presents tokens containing the normalized RGB color values and relative pixel coordinates only for the entities (max 8 tokens), while the other presents a token for each pixel plus the key currently held by the agent (50 tokens). Like 4.3.2, the underlying learning dynamics are the same, but the pixel observations contain a significantly higher volume and ratio of noise. We use 5 seeds per method and train for 40 million steps. Additional training parameters can be found in the appendix C.6.

In Figure 8, we observe that MaxPool is able to achieve superior sample efficiency and performance on the entity-level observations, but then collapses to the worst performance under pixel-level observations. As in the previous experiment, we observe the smallest decline from AdaPool, which achieves superior performance in the high noise regime. AvgPool and ClsToken perform comparatively worse in both regimes. Regarding the initial high performance of MaxPool, the goal gem is always represented by a white pixel, giving it a normalized color value of all ones – the upper bound of the observation space. Corollary 3.8 implies that MaxPool is uniquely suited for problems of this form, hence the superior performance in 10a. However, the non-entity pixels (empty space) are a light grey color, relatively close in numerical representation to white. This likely exacerbated the performance drop, despite MaxPool's helpful inductive bias. Overall, this demonstrates that our theoretical framework can be used as an analytical tool for neural network designers to map their problem domains to the best pooling methods for the job, rather than relying on intuition or trial and error.

## 4.5. CIFAR

Prior experiments carefully controlled for signal and noise, but left the following unanswered: (1) Does our analysis hold up on real-world data where signal and noise are less clearly defined, and where individual vectors may contain a mix of both? (2) How do you employ AdaPool when the choice of query is not obvious? To answer these questions, we conducted additional studies on image classification using the CIFAR 10 and 100 benchmark datasets. We adopted the Vision Transformer (ViT) approach (Dosovitskiy et al., 2021), partitioning the 32x32 pixel RGB images into 64 separate 4x4 pixel patches as shown in Figure 9. These are then flattened, projected, and fed into the transformer for embedding. Standard ViT implementations use ClsToken to pool output embeddings, with a minority favoring AvgPool.

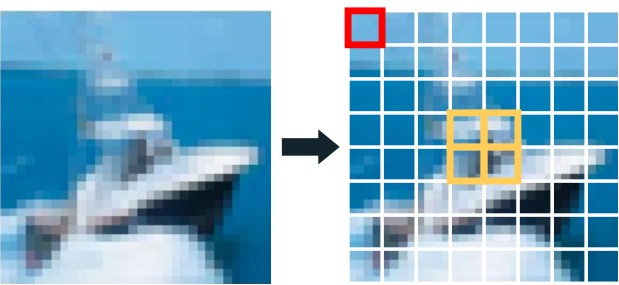

Figure 9: Splitting the CIFAR images into patches. We experiment with different choices of query for AdaPool, with the *Corner* query using the red patch embedding, the *Focal* query averaging the four central yellow patch embeddings, and the *Mean* query averaging all patch embeddings.

Most samples in the CIFAR dataset tend to contain the object being classified in the center of the image. While periphery patches may include some useful information, like grass or water, we hypothesize that the central patches are more likely to contain discriminating signal on average than any other patches. We thus examine 3 choices of query: *Corner*, the embedding of the upper-left-most periphery patch; *Focal*, averaging the embeddings of the four central patches; and *Mean*, using the average of all patch embeddings as the query. These query choices are also highlighted in Figure 9.

We report the Top-1 accuracy scores on the holdout test set in Table 1, averaged from the best models across 5 folds after 300 training epochs each; see additional details in C.7. We observe that AdaPool *Focal* and *Mean* queries outperform all other methods in both cases, reaching similar accuracies. The *Corner* query underperforms in both experiments, aligning with the prediction that a noisy query should lead to worse performance. By evaluating the efficacy of different queries, one could discover how signal tends to be distributed. This evidence supports the hypothesis that signal tends to be more concentrated in the central patches.

| METHOD | CIFAR-10 | CIFAR-100 |
|---|---|---|
| ClsToken | 84.52 $\pm 0.21$ | 55.56 $\pm 0.13$ |
| AvgPool | 87.15 $\pm 0.35$ | 59.63 $\pm 0.23$ |
| MaxPool | 87.65 $\pm 0.17$ | 60.55 $\pm 0.28$ |
| Ada-*Focal* | **87.98** $\pm 0.42$ | **61.22** $\pm 0.33$ |
| Ada-*Mean* | 87.84 $\pm 0.30$ | **61.23** $\pm 0.20$ |
| Ada-*Corner* | 87.00 $\pm 0.30$ | 57.08 $\pm 0.31$ |

Table 1: ViT Top-1 test accuracy on the CIFAR image classification dataset using different pooling methods.

Additionally, the standard ClsToken approach underperforms all other methods by a significant margin. AvgPool and MaxPool perform relatively well, with MaxPool having the edge in both experiments. In Figure 4, we previously observed that this occurred in the very low SNR regime (SNR < 0.1). If that trend generalizes beyond our synthetic dataset, then it is a quantitative indication that, after embedding, CIFAR samples tend to be signal-sparse.

One of our major theoretical results was that the selection of query is critical for approximating the optimal quantizer. This could be a practical limitation, depending on AdaPool's sensitivity to non-ideal queries. A key takeaway from this experiment is that, while AdaPool is indeed sensitive to poor choices like the *Corner* query, the *Mean* query is a strong default option that can outperform the standard pooling methods, even if the data is more noise than signal.

## 5. Conclusion

In this work, we investigated the selection and design of global pooling methods for aggregating embeddings produced by transformers. We drew connections between pooling, vector quantization, and associative memory to reframe pooling as a lossy compression problem rather than a trivial operation for aligning dimensions. This reframing helped formalize the limitations of common approaches like Avg-Pool and MaxPool, and we showed that attention-based *adaptive pooling*, a niche approach, approximates the optimal compressor within derived bounds. These theoretical findings were first evaluated with carefully designed supervised experiments, and then in more general reinforcement learning and vision tasks using the Multi-Particle Environment, BoxWorld, and CIFAR. The results confirmed that AvgPool and MaxPool fail in predictable ways when the signal-to-noise ratio of inputs changes, validating the theoretical analysis. Additionally, ClsToken had a similar noise sensitivity to AvgPool, with less predictable relative performance between tasks. Crucially, we found that AdaPool was predictably robust to different ratios and quantities of noise, resulting in superior performance across tasks.

## Acknowledgements

We thank Mark Fleischer, Joshua McClellan, Zachary Goddard, John Winder, and Jovanna Aragon for their helpful feedback.

## Impact Statement

The underlying motivation of this paper is to enhance the design of neural network architectures used for controlling autonomous systems. We focus on formalizing and improving one particular aspect, vector pooling, which is a commonly used and broadly applicable technique in many machine learning domains. In terms of specific impacts, there is a rich body of discussion on the very real potential benefits and harms of intelligent autonomous systems, none of which we feel merits particular discussion in this work.

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

# A. Proofs

## A.1. Proof of Corollary 3.3

The point $\mathbf{x}_c^*$ that minimizes signal loss is the centroid of the signal subset $\mathbf{X}_s \subseteq \mathbf{X}$.

*Proof.* Given a vector set $\mathbf{X}$ as presented in Section 3.1, a global vector pool $C$ yielding $C(\mathbf{X}) = \mathbf{x}_c \in \mathbb{R}^d$, and the following definition for signal loss given in Section 3.2

$$\mathbb{L}(\mathbf{X}_s, \mathbf{x}_c) = \frac{1}{k} \sum_{\mathbf{x}_i \in \mathbf{X}_s} (\mathbf{x_i} - \mathbf{x}_c))^2$$

the signal-loss minimizing point $\mathbf{x}_c^*$ is given by

$$\frac{\partial \mathbb{L}}{\partial \mathbf{x}_c} = -\frac{2}{k} \sum_{\mathbf{x}_i \in \mathbf{X}_s} (\mathbf{x_i} - \mathbf{x}_c^*) = 0$$

$$-k \cdot \mathbf{x}_c^* + \sum_{\mathbf{x}_i \in \mathbf{X}_s} \mathbf{x_i} = 0$$

$$\mathbf{x}_c^* = \frac{1}{k} \sum_{\mathbf{x}_i \in \mathbf{X}_s} \mathbf{x_i}$$

yielding the centroid of the signal subset. $\quad\square$

## A.2. Proof of Corollary 3.5

AvgPool is an optimal vector compressor if the input set contains no noise.

$$\mathbf{X}_\eta = \varnothing \implies AvgPool = C^*$$

*Proof.* Suppose $\mathbf{X}_\eta = \varnothing$. Then $\mathbf{X}_s = \mathbf{X}$, $k = N$, and

$$AvgPool(\mathbf{X}) = \frac{1}{N} \sum_{\mathbf{x}_i \in \mathbf{X}} \mathbf{x_i} = \frac{1}{k} \sum_{\mathbf{x}_i \in \mathbf{X}_s} \mathbf{x_i} = C^*(\mathbf{X})$$

$$\square$$

## A.3. Proof of Corollary 3.6

When the input contains noise, $\mathbf{X}_\eta \neq \varnothing$, AvgPool is an optimal vector compressor if and only if the centroid of $X_s$ is equivalent to the centroid of $X_\eta$.

$$AvgPool = C^* \iff$$

$$AvgPool(\mathbf{X}_s) = AvgPool(\mathbf{X}_\eta)$$

*Proof.* **Case 1.** ($\implies$) Suppose $\mathbf{X}_\eta \neq \varnothing$ and $AvgPool = C^*$. Since AvgPool assigns weight $w_i = \frac{1}{N}$ to all vectors while the optimal compressor $C^*$ assigns weights $w_i = \frac{1}{k}$ to all $\mathbf{x}_s$ and $w_i = 0$ to all $\mathbf{x}_\eta$, we have

$$AvgPool = C^* \implies$$

$$\frac{1}{N} \sum_{\mathbf{x}_i \in \mathbf{X}_s} \mathbf{x_i} + \frac{1}{N} \sum_{\mathbf{x}_i \in \mathbf{X}_\eta} \mathbf{x_i} = \frac{1}{k} \sum_{\mathbf{x}_i \in \mathbf{X}_s} \mathbf{x_i}$$

$$\frac{1}{N} \sum_{\mathbf{x}_i \in \mathbf{X}_\eta} \mathbf{x_i} = \left(\frac{1}{k} - \frac{1}{N}\right) \sum_{\mathbf{x}_i \in \mathbf{X}_s} \mathbf{x_i}$$

$$\frac{1}{N} \sum_{\mathbf{x}_i \in \mathbf{X}_\eta} \mathbf{x_i} = \frac{N-k}{N} \cdot \frac{1}{k} \sum_{\mathbf{x}_i \in \mathbf{X}_s} \mathbf{x_i}$$

$$\frac{1}{N-k} \sum_{\mathbf{x}_i \in \mathbf{X}_\eta} \mathbf{x_i} = \frac{1}{k} \sum_{\mathbf{x}_i \in \mathbf{X}_s} \mathbf{x_i}$$

$$AvgPool(\mathbf{X}_\eta) = AvgPool(\mathbf{X}_s)$$

**Case 2.** ($\impliedby$) Suppose $\mathbf{X}_\eta \neq \varnothing$ and $AvgPool(\mathbf{X}_s) = AvgPool(\mathbf{X}_\eta)$. Then

$$AvgPool(\mathbf{X}_s) = AvgPool(\mathbf{X}_\eta) \implies$$

$$\frac{1}{k} \sum_{\mathbf{x}_i \in \mathbf{X}_s} \mathbf{x_i} = \frac{1}{N-k} \sum_{\mathbf{x}_i \in \mathbf{X}_\eta} \mathbf{x_i}$$

The remainder of the proof is symmetric to Case 1. $\quad\square$

## A.4. Proof of Corollary 3.8

MaxPool is signal-optimal if and only if the input set $\mathbf{X}$ contains a single signal vector ($k = 1$) taking the maximal value along each feature dimension.

$$MaxPool = C^* \iff |\mathbf{X}_s| = 1, MaxPool(\mathbf{X}) = \mathbf{x}_s$$

*Proof.* **Case 1.** ($\implies$) Suppose $MaxPool = C^*$. When $\mathbf{x}_i \in \mathbf{X}_s$ is a signal vector, the weight vector yielded by MaxPool is $\mathbf{w}_i = [w_{i,1}^*, \ldots, w_{i,d}^*] = \mathbf{w}_i^*$, where $w_{i,d}^* = \frac{1}{k}$ for all features $d$.

Since the weights assigned by MaxPool can only take values of 1 or 0, this implies $w_{i,d}^* = 1$ for all features $d$ which, in turn, implies that $k = |\mathbf{X}_s| = 1$. Furthermore, we know that MaxPool only assigns a weight of 1 when $x_{i,d}$ takes the max value over the set along a dimension $d$. Since $\mathbf{w}_i = [1, ..., 1]$, it implies that $\mathbf{x}_i = MaxPool(\mathbf{X})$. By our assumption, $\mathbf{x}_i = \mathbf{x}_s = MaxPool(\mathbf{X})$.

**Case 2.** ($\impliedby$) Suppose $|\mathbf{X}_s| = 1, MaxPool(\mathbf{X}) = \mathbf{x}_s$. When $\mathbf{x}_i \in \mathbf{X}_s$ is the only signal vector, by definition of MaxPool, the weight vector must be $\mathbf{w}_i = [w_{i,1}, \ldots, w_{i,d}]$ where $w_{i,d}^* = 1$ for all features. By our assumption, $|\mathbf{X}_s| = 1 = k$, so the optimal weight vector and the MaxPool weight vector coincide, $\mathbf{w}_i = \mathbf{w}_i^*$.

Now instead, when $\mathbf{x}_i \in \mathbf{X}_\eta$ is a noise vector. Since $MaxPool(\mathbf{X}) = \mathbf{x}_s$, by definition, MaxPool assigns a weight of $w_{i,d} = 0$ for all features of all noise vectors,

which coincides with the weight vector yielded by the signal-optimal global vector pool $\mathbf{w}_i^* = [0, \ldots, 0]$. Since Max-Pool assigns the optimal weights to all vectors in the $\mathbf{X}$, $MaxPool = C^*$.

$\square$

## A.5. Proof of Theorem 3.12

For any signal-to-noise ratio, AdaPool can approximate the signal-optimal vector pool within an error bound determined by the distribution of signal and noise relations.

Explicitly, for any signal vector $x_i \in X_s$, AdaPool approximates the optimal weight of $w_i^* = \frac{1}{k}$ within error bounds

$$L \leq w_i^* - w_i \leq U$$

where L and U are the following lower and upper bounds respectively:

$$L = \frac{1}{k} - \left(1 + (k-1) \cdot e^{-\epsilon_s} + (N-k) \cdot e^{-D}\right)^{-1}$$

$$U = \frac{1}{k} - \left(1 + (k-1) \cdot e^{\epsilon_s} + (N-k) \cdot e^{-M}\right)^{-1}$$

and for any noise vector $x_i \in X_\eta$, AdaPool approximates the optimal weight of $w_i^* = 0$ with error bound

$$L \leq w_i^* - w_i \leq U$$

where L and U are the following lower and upper bounds respectively:

$$L = -\left(k \cdot e^M + 1 + (N-k-1) \cdot e^{-\epsilon_\eta}\right)^{-1}$$

$$U = -\left(k \cdot e^D + 1 + (N-k-1) \cdot e^{\epsilon_\eta}\right)^{-1}$$

*Proof.* First, we let $W_V = I_d$. We then start by expanding the equation for the attention weight corresponding to $\mathbf{x}_i$ as follows:

$$w_i = \frac{e^{r_i}}{\sum\limits_{j \in \mathbf{X}} e^{r_j}}$$

$$= \frac{1}{\sum\limits_{j \in \mathbf{X}} e^{r_j - r_i}}$$

$$= \left(\sum_{s \in \mathbf{X}_s} e^{r_s - r_i} + \sum_{\eta \in \mathbf{X}_\eta} e^{r_\eta - r_i}\right)^{-1}$$

As a general technique, we substitute the maximal and minimal neighborhood sizes well as maximal and minimal margins between signal and noise relations as defined in Section 3.5 to bound the range of the softmax operation.

**Case 1.** *Weight for a signal vector*: Given $\mathbf{x}_i \in \mathbf{X}_s$, it follows that $r_i$ belongs to the set of signal relations $\{r_s\}$. We can obtain an upper bound on the corresponding weight $w_i$ via

$$w_i = \left(1 + \sum_{s \neq i} e^{r_s - r_i} + \sum_\eta e^{r_\eta - r_i}\right)^{-1}$$

$$\leq \left(1 + (k-1) \cdot e^{-\epsilon_s} + \sum_\eta e^{r_\eta - r_i}\right)^{-1}$$

$$\leq \left(1 + (k-1) \cdot e^{-\epsilon_s} + (N-k) \cdot e^{-D}\right)^{-1}$$

$$= w_s^U$$

Intuitively, we use the lower bound distance $-\epsilon_s$ between any two signal relation values along with the maximal margin $D = M + \epsilon_s + \epsilon_\eta$ between signal and noise to bound the value for $w_i$. We can similarly obtain a lower bound on $w_i$ with the following:

$$w_i = \left(1 + \sum_{s \neq i} e^{r_s - r_i} + \sum_\eta e^{r_\eta - r_i}\right)^{-1}$$

$$\geq \left(1 + (k-1) \cdot e^{\epsilon_s} + \sum_\eta e^{r_\eta - r_i}\right)^{-1}$$

$$\geq \left(1 + (k-1) \cdot e^{\epsilon_s} + (N-k) \cdot e^{-M}\right)^{-1}$$

$$= w_s^L$$

In this case, the largest possible distance $+\epsilon_s$ between any two signal relation values and the minimum margin $M$ are used. Note that these bounds are independent of any particular sample from the input set, instead relying purely on characteristics of the input distributions defined in the assumptions.

With these bounds on $w_i$, it follows that the upper and lower bounds on the error from the optimal weight $w_i^* = \frac{1}{k}$ are

$$L = \frac{1}{k} - w_s^U \leq w_i^* - w_i \leq \frac{1}{k} - w_s^L = U$$

**Case 2.** *Weight for a noise vector*: Given $\mathbf{x}_i \in \mathbf{X}_\eta$, it follows that $r_i$ belongs to the set of noise relations $\{r_\eta\}$. We can obtain an upper bound on the corresponding weight

$w_i$ via

$$w_i = \left( \sum_s e^{r_s - r_i} + 1 + \sum_{\eta \neq i} e^{r_\eta - r_i} \right)^{-1}$$

$$\leq \left( k \cdot e^M + 1 + \sum_\eta e^{r_\eta - r_i} \right)^{-1}$$

$$\leq \left( k \cdot e^M + 1 + (N - k - 1) \cdot e^{-\epsilon_\eta} \right)^{-1}$$

$$= w_\eta^U$$

Similar to Case 1, we use the lower bound distance $-\epsilon_s$ between any two noise relationship scores along with the minimum margin $M$ between signal and noise to the value for $w_i$ from above. We can similarly obtain a lower bound on $w_i$ with the following:

$$w_i = \left( \sum_s e^{r_s - r_i} + 1 + \sum_{\eta \neq i} e^{r_\eta - r_i} \right)^{-1}$$

$$\geq \left( k \cdot e^D + 1 + \sum_\eta e^{r_\eta - r_i} \right)^{-1}$$

$$\geq \left( k \cdot e^D + 1 + (N - k - 1) \cdot e^{\epsilon_\eta} \right)^{-1}$$

$$= w_\eta^L$$

Again, the largest possible distance $+\epsilon_s$ between any two signal relationship scores and the maximum margin $D$ are used to bound $w_i$ from below.

With these bounds on $w_i$, it follows that the upper and lower bounds on the error from the optimal weight $w_i^* = 0$ are

$$L = 0 - w_s^U \leq w_i^* - w_i \leq 0 - w_s^L = U$$

$\square$

### A.6. Proof of Corollary 3.10

AvgPool is a special case of AdaPool.

*Proof.* We will show this by construction. Let $W_Q = 0_d$ be a zero matrix, then the formula for the weights reduces to

$$w_i = \frac{\exp(0)}{\sum_j^N \exp(0)} = \frac{1}{N}$$

By letting $W_V = I_d$ be the identity, the formula for AdaPool then becomes and

$$AdaPool(\mathbf{X}) = \sum_i^N w_i \cdot \mathbf{x}_i = AvgPool(\mathbf{X})$$

$\square$

### A.7. Proof of Corollary 3.11

MaxPool is a special case of AdaPool.

*Note:* The following proof assumes a multi-head implementation of AdaPool (attention), which was not discussed in the main text for brevity. We outline the details of multi-head attention in B.2.

*Proof.* We will show this by construction. First, we assume a multihead adaptive pooling mechanism with as many heads as input dimensions, or $h = d$. This is done to compute a unique attention weight for each feature of each vector. Let $W_K, W_V = I_d$, and let $W_Q = a \cdot I_d$ be the identity matrix scaled by some large constant $a$. As $a \to \infty$, the softmax of a 1-dimensional head approaches the maximum over some column $d$:

$$\lim_{a \to \infty} w_{i,d} = \frac{\exp\left( \frac{a}{\sqrt{d}} \cdot x_{q,d} \cdot x_{i,d} \right)}{\sum_j^N \exp\left( \frac{a}{\sqrt{d}} \cdot x_{q,d} \cdot x_{j,d} \right)}$$

$$= \frac{\exp(\beta \cdot x_{i,d})}{\sum_j^N \exp(\beta \cdot x_{j,d})}$$

$$= \begin{cases} 1 & \text{if } x_{i,d} > x_{j,d}, \forall j \in N \\ 0 & \text{otherwise} \end{cases}$$

where we substitute $\beta = \frac{a}{\sqrt{d}} \cdot x_{q,d}$. By then concatenating all of the heads, we obtain weight vectors $\mathbf{w}_i \in \{0, 1\}^d$ corresponding to each input vector $\mathbf{x}_i$, and the formula for AdaPool then becomes

$$AdaPool(\mathbf{X}) = \sum_i^N \mathbf{w}_i \cdot \mathbf{x}_i = MaxPool(\mathbf{X})$$

$\square$

## B. Additional Results

### B.1. Time Complexity of AdaPool

AvgPool and MaxPool can be implemented with $O(n \cdot d)$ algorithms, as both require each of the $d$ features of each of the $n$ vectors to be visited during aggregation.

Self-attention famously experiences quadratic computational costs with respect to the context size $n$. In particular, the attention weight computation is $O(n^2 \cdot d)$, as each vector in the input set computes attention weights for each other vector in the input set, i.e. $n^2$ dot products of $d$ dimensional vectors. However, AdaPool uses cross-attention with a single query, requiring the computation of only a single set of attention weights rather than $n$ sets. This reduces the weight computation and pooling time down to $O(n \cdot d)$, the same as AvgPool and MaxPool.

However, AdaPool still retains the overhead of the $QKV$ projections of the input set, which involves standard matrix multiplications for each input vector, running $O(n \cdot d^2)$. Thus, the overall time complexity of AdaPool is $O(n \cdot d + n \cdot d^2)$. The key takeaway is that it scales linearly with the number of inputs but quadratically with the number of features (like a standard linear layer).

In isolation, AdaPool is slower – the increased expressivity comes at a computational cost. However, in practice, these pooling layers are placed at the end of a multi-layer transformer or similarly sized encoder network, and the compute time added by the pooling layer is marginal. For our particular applications, when using a transformer with 3+ layers, the time difference between AdaPool and Avg/MaxPool networks was negligible, regardless of the size of $d$.

Also of note, the ClsToken implementation is significantly slower than all other methods. This is due to the fact that the learned class embedding must be copied and concatenated along the batch dimension of the input for every inference, which is quite expensive in terms of both time and memory during training. This is a known hindrance investigated by Zhai et al. (2022), which led them to explore the use of AvgPool for efficiency reasons when scaling vision transformers.

### B.2. Multi-Head AdaPool

In the theoretical portions of this work, we primarily considered the use of a single-headed attention for simplicity and conciseness of notation and proofs. However, we note that implementing multi-head attention, as proposed by Vaswani et al. (2017), is quite straightforward. This can be obtained by partitioning the input vector space $\mathbb{R}^d$ into $h$ partitions (i.e. heads) belonging to $\mathbb{R}^{d/h}$, computing a separate set of adaptive weights over the inputs per head, and then concatenating the resulting weighted averages from each head. We assume this form of AdaPool in Proof A.7.

Multi-head adaptive pooling presents one key potential benefit that we did not mention in the main text – it allows different subspaces of the input to be pooled in parallel. This means that, with further work, our analysis could potentially be generalized beyond vectors that purely signal or purely noise, more akin to what we expect from real-world data. Consider a vector that contains meaningful signal in one subspace, but serves as noise in another. In an autonomous system, one could imagine that a faulty sensor providing accurate position but bad velocity measurements might create such an input. By using multihead attention with the right partitioning, AdaPool could assign it a high weight in the position subspace while suppressing it in the velocity subspace, where it is noisy. While this multi-head partitioning allows us to relax some of the strict assumptions made about the input data, it does place more relative

burden on the query, ideally requiring it to contain signal in all subspaces used in downstream processing.

### B.3. Aggregation Function Approximation

Before we evaluate the noise robustness problem, we first ask the following question: *Does the choice of output pooling even matter when the transformer is composed of a bunch of learned weighted averaging (attention) layers?* As a simple test, we examine how well a transformer with each pooling method can approximate each other. Using the synthetic dataset, we generate 3 sets of targets: $y = MaxPool(X)$, $y = MinPool(X)$, and $y = AvgPool(X)$. If the pooling method was of little consequence, we would expect all models to achieve the same performance regardless of the method used.

| METHOD | MAX | MEAN | MIN |
|---|---|---|---|
| MaxPool | **0.000**
±0.000 | 0.026
±0.046 | 0.297
±0.239 |
| AvgPool | 0.003
±0.000 | **0.000**
±0.000 | 0.003
±0.000 |
| ClsToken | 0.011
±0.000 | 0.008
±0.000 | 0.011
±0.000 |
| AdaPool | 0.002
±0.000 | **0.000**
±0.000 | **0.002**
±0.000 |

Table 2: Aggregation Approximation MSE (N=128, d=16)

Unsurprisingly, pooling methods can approximate themselves perfectly, with MaxPool achieving near-zero MSE on the max targets and AvgPool achieving near-zero MSE on the average targets. Interestingly, MaxPool performance deteriorates on the average and min targets. AdaPool is also able to perfectly fit the average targets and has the best performance on min targets, with a similarly good performance on max targets. AvgPool performs just marginally worse than AdaPool on min and max targets, while ClsToken consistently performs worse than AdaPool and AvgPool on all targets. This shows that the inductive bias of the pooling method chosen has a clear impact on a transformer encoder's ability to approximate functions. It is also worth noting that, although we have proven MaxPool to be a special case of AdaPool, AdaPool is not able to perfectly fit the max targets. This is likely because the approximation requires that one of the weight matrices obtain extremely high values, which is unlikely to occur in practice via gradient descent.

### B.4. Noise Robustness Table

We provide the data for Figure 4 in Table 3. This corresponds to the signal loss of each method for different signal-to-noise ratios of the input data.

| METHOD | KNN-1 *SNR = 0.01* | KNN-2 *SNR = 0.02* | KNN-4 *SNR = 0.03* | KNN-8 *SNR = 0.06* | KNN-16 *SNR = 0.13* | KNN-32 *SNR = 0.25* | KNN-64 *SNR = 0.50* | KNN-128 *SNR = 1.00* |
|---|---|---|---|---|---|---|---|---|
| MaxPool | 0.066 ±0.102 | 0.061 ±0.110 | 0.050 ±0.093 | 0.045 ±0.089 | 0.047 ±0.095 | 0.055 ±0.096 | 0.068 ±0.096 | 0.046 ±0.081 |
| AvgPool | 0.093 ±0.000 | 0.071 ±0.000 | 0.055 ±0.000 | 0.043 ±0.000 | 0.031 ±0.000 | 0.016 ±0.008 | **0.001** ±0.000 | **0.000** ±0.000 |
| ClsToken | 0.102 ±0.000 | 0.079 ±0.000 | 0.063 ±0.000 | 0.050 ±0.000 | 0.038 ±0.000 | 0.027 ±0.000 | 0.016 ±0.000 | 0.008 ±0.000 |
| AdaPool | **0.037** ±0.006 | **0.019** ±0.005 | **0.006** ±0.001 | **0.004** ±0.000 | **0.003** ±0.000 | **0.002** ±0.000 | **0.001** ±0.000 | **0.000** ±0.000 |
| Baseline *(Centroid)* | 0.093 | 0.071 | 0.055 | 0.043 | 0.031 | 0.020 | 0.008 | 0.000 |
| Baseline *(Target)* | 0.058 | 0.044 | 0.040 | 0.041 | 0.048 | 0.060 | 0.080 | 0.126 |

Table 3: Signal Loss (MSE) on the KNN-Centroid Task (N=128, d=16) as presented in Figure 4.

## B.5. KNN Centroid Ablations

We performed several ablation studies to examine how key data and network hyperparameters changed the relative performance of each pooling method. The ablations were performed on much smaller cardinality datasets than the main experiment to enable faster iteration. We perform an initial study to use as a baseline with which we compare each ablation, using a synthetic dataset with 1 million samples, each of size N=32 and d=16. We use the exact same network architecture as the main experiment, with 12 layers and a hidden dimension of 16. We perform evaluations using k-fold cross-validation on the KNN-Centroid task as described in section 4.2, and we report results for the baseline in Table 4. The results reflect the main experiment, with AdaPool exhibiting superior performance across different SNRs on a dataset with samples that have a quarter of the cardinality as those in the main experiment (32 vs 128), with all other hyperparameters being the same.

Next, we examine the effect of increasing the dimensionality of the input vectors from d=16 to d=64. As shown in Table 5, MaxPool slightly outperforms AdaPool in the low signal regimes, while AdaPool outperforms the other methods in the medium to high signal regimes. AvgPool and ClsToken perform very poorly in the low signal regime, with AvgPool getting better as signal increases much more rapidly than ClsToken.

We then look at expanding the width of the network from a hidden dimension of 16 to 32 to 64. As displayed in tables 8 and 7, AdaPool outperforms all methods at all SNRs for all widths. However, we observe that the absolute signal loss for all methods reduces as the network width increases. In particular, AvgPool and MaxPool become much more competitive in the higher signal regimes. One possible interpretation of this phenomenon is that as the width of

the network grows relative to the input data dimension, the burden on the pooling mechanism to compress information is relaxed.

Finally, we look at increasing the depth of the network from 12 layers to 24 layers. The results in Table 8 do not show a clearly dominant method in any particular SNR regime. AdaPool appears to be most consistent, but is edged out by AvgPool in the high signal regimes. AvgPool does predictably poorly in the high noise regime, while MaxPool does predictably poorly in the high signal regime. Finally, ClsToken performs the worst by far in all regimes.

## C. Experiment Parameters

### C.1. Synthetic Dataset Construction Parameters

To construct a dataset full of sets of points with diverse distributions, we use the following sampling scheme. For each of the $d$ features, all $N$ vectors sample values from one of the following distributions:

Gaussian with randomly sampled mean and standard deviation:

$$\mu \sim Uniform(-3, 3)$$
$$\sigma \sim Uniform(1, 3)$$
$$x_{:,d} \sim Normal(\mu, \sigma)$$

Exponential with randomly sampled rate parameter, sign, and shift:

$$sign \sim \{-1, 1\}$$
$$shift \sim Uniform(0, 3) * sign$$
$$\lambda \sim Uniform(0.1, 2)$$
$$x_{:,d} \sim Exponential(\lambda) * sign - shift$$

| $N$ | $d$ | $hid$ | $L$ | METHOD | SNR $\frac{1}{32}$ | SNR $\frac{2}{32}$ | SNR $\frac{4}{32}$ | SNR $\frac{8}{32}$ | SNR $\frac{16}{32}$ |
|---|---|---|---|---|---|---|---|---|---|
| 32 | 16 | 16 | 12 | MaxPool | 0.022 | 0.014 | 0.009 | 0.006 | 0.009 |
| | | | | AvgPool | 0.086 | 0.049 | 0.010 | 0.005 | 0.003 |
| | | | | ClsToken | 0.094 | 0.066 | 0.046 | 0.030 | 0.017 |
| | | | | AdaPool | **0.018** | **0.011** | **0.007** | **0.004** | **0.002** |

Table 4: Ablation baseline study.

| $N$ | $d$ | $hid$ | $L$ | METHOD | SNR $\frac{1}{32}$ | SNR $\frac{2}{32}$ | SNR $\frac{4}{32}$ | SNR $\frac{8}{32}$ | SNR $\frac{16}{32}$ |
|---|---|---|---|---|---|---|---|---|---|
| 32 | **64** | 16 | 12 | MaxPool | **0.0056** | **0.0025** | 0.0025 | 0.0018 | 0.0018 |
| | | | | AvgPool | 0.0204 | 0.0104 | 0.0053 | 0.0023 | 0.0006 |
| | | | | ClsToken | 0.0229 | 0.0126 | 0.0069 | 0.0036 | 0.0017 |
| | | | | AdaPool | 0.0064 | 0.0031 | **0.0010** | **0.0005** | **0.0002** |

Table 5: Ablation increasing the data dimension $\times 4$ from 16 to 64.

| $N$ | $d$ | $hid$ | $L$ | METHOD | SNR $\frac{1}{32}$ | SNR $\frac{2}{32}$ | SNR $\frac{4}{32}$ | SNR $\frac{8}{32}$ | SNR $\frac{16}{32}$ |
|---|---|---|---|---|---|---|---|---|---|
| 32 | 16 | **32** | 12 | MaxPool | 0.022 | 0.011 | 0.0023 | 0.0006 | 0.0003 |
| | | | | AvgPool | 0.048 | 0.022 | 0.0012 | 0.0006 | 0.0002 |
| | | | | ClsToken | 0.086 | 0.057 | 0.0371 | 0.0214 | 0.0082 |
| | | | | AdaPool | **0.003** | **0.001** | **0.0006** | **0.0003** | **0.0001** |

Table 6: Ablation increasing the network width $\times 2$ from 16 to 32.

| $N$ | $d$ | $hid$ | $L$ | METHOD | SNR $\frac{1}{32}$ | SNR $\frac{2}{32}$ | SNR $\frac{4}{32}$ | SNR $\frac{8}{32}$ | SNR $\frac{16}{32}$ |
|---|---|---|---|---|---|---|---|---|---|
| 32 | 16 | **64** | 12 | MaxPool | 0.022 | 0.0012 | 0.0005 | 0.00024 | 0.00010 |
| | | | | AvgPool | 0.004 | 0.0016 | 0.0005 | 0.00021 | 0.00009 |
| | | | | ClsToken | 0.086 | 0.0574 | 0.0373 | 0.02147 | 0.00825 |
| | | | | AdaPool | **0.001** | **0.0010** | **0.0003** | **0.00019** | **0.00007** |

Table 7: Ablation increasing the network width $\times 4$ from 16 to 64.

| $N$ | $d$ | $hid$ | $L$ | METHOD | SNR $\frac{1}{32}$ | SNR $\frac{2}{32}$ | SNR $\frac{4}{32}$ | SNR $\frac{8}{32}$ | SNR $\frac{16}{32}$ |
|---|---|---|---|---|---|---|---|---|---|
| 32 | 16 | 16 | **28** | MaxPool | 0.020 | **0.011** | 0.007 | 0.005 | 0.0056 |
| | | | | AvgPool | 0.061 | 0.013 | **0.006** | **0.003** | **0.0019** |
| | | | | ClsToken | 0.094 | 0.066 | 0.046 | 0.030 | 0.0165 |
| | | | | AdaPool | **0.017** | 0.014 | 0.014 | 0.008 | 0.0021 |

Table 8: Ablation increasing the network depth $\times 2$ from 12 to 24.

Uniform with randomly sampled lows and highs:

$low \sim Uniform(-3, 3)$
$high \sim Uniform(0.2, 3) + low$
$x_{:,d} \sim Uniform(low, high)$

We scale each resulting vector by $\sqrt{d}$ to reduce the impact of the curse of dimensionality on the scale of the loss when different dimensionalities are used. The features for a given set are drawn evenly from these 3 distributions, with each individual feature having a uniquely parametrized distribution according to the above scheme. The ordering of these

feature distributions is shuffled randomly for each sample. For our main supervised experiment, we generate 1 million samples, 128 vectors per sample, and 16 features per vector. Our dataset was generated using NumPy version 2.0.2 with a seed of 42.

## C.2. Supervised KNN Centroid

The base network consisted of a transformer encoder with 12 layers, a hidden dimension of 16 (same as the input data dimension), a feedforward dimension of 64, and 8 attention heads. We follow the standard practice of moving the layer norm to the input of the attention and feedforward sublayers (pre-norm). We use PyTorch for all model implementations. Weight matrices and embedding vectors are initialized according to $Normal(\mu = 0.0, \sigma = 0.02)$ with bias set to 0.0. Layer norms are initialized with weight set to 1.0 and bias set to 0.0.

### C.2.1. MODEL ARCHITECTURE

| Hyperparameters | Value |
|---|---|
| Input Shape | $N \times 16$ |
| Num Layers | 12 |
| Num Heads | 8 |
| Dim Hidden | 16 |
| Dim FF | 64 |
| Dropout Attn | 0.0 |
| Dropout FF | 0.1 |
| Bias Attn | False |
| Bias FF | True |

Table 9: Network Hyperparameters for KNN-Centroid

We train on the main dataset described above (N=128, d=16). For each desired level of noise, we generated **X**,**y** pairs by picking an arbitrary target vector from the input set **X**, finding the k-nearest neighbors to that target point, and computing **y** to be the centroid of those $k$ neighbors (excluding the target). These k neighbors are considered the signal vectors, while the rest are considered noise. We add a learned embedding to the target vector at inference time to indicate it to the transformer. We compute **X**,**y** pairs for the following values of $k : 1, 2, 4, 8, 16, 32, 64, 128$. For each of these pairs, we use 5-fold cross-validation with a holdout test set of 100k samples. Each transformer + pooling method (AdaPool, ClsToken, AvgPool, MaxPool) was trained for 100 epochs with a learning rate of $5.0e - 4$ and a batch size of 750. The base transformer model is initialized with the same weights for each method, and different seeds are used on each fold.

## C.3. Supervised Aggregation Approximation

We use the same architecture and training setup outlined in the previous section. Given the main dataset generated above according to C.1, we produced three sets of targets $y$ using standard pooling functions to see if the underlying transformer was good enough to approximate all aggregation types regardless of the method used to aggregate its outputs. Targets include $y = MinPool(X)$, $y = MaxPool(X)$, and $y = AvgPool(X)$.

## C.4. MPE Simple Centroid

For the multi-agent experiments on the simple centroid task, network architecture hyperparameters are displayed in Table 10 and training hyperparameters in Table 11.

| Hyperparameters | Value |
|---|---|
| Obs Shape | $N \times 8$ |
| Num Layers | 3 |
| Num Heads | 8 |
| Dim Input | 8 |
| Dim Hidden | 8 |
| Dim FF | 32 |
| Dropout Attn | 0.0 |
| Dropout FF | 0.1 |
| Bias Attn | False |
| Bias FF | True |
| Action MLP Layers | 3 |
| Action MLP Dim | 32 |
| Value MLP Layers | 3 |
| Value MLP Dim | 32 |
| Continuous Actions | True |

Table 10: Network Hyperparameters for MPE Simple Centroid

| Hyperparameters | Value |
|---|---|
| Algorithm | PPO |
| Number of Seeds | 10 |
| Number of Episodes | 500 |
| Training Batch | 8192 Agent Steps |
| Minibatch | 8192 Agent Steps |
| SGD Iter/Batch | 16 |
| Episode Length | 128 Env Steps |
| Learning Rate | $2.5e - 3$ |
| $\gamma$ | 0.99 |
| $\lambda$ | 0.95 |
| KL Coeff | 0.0 |
| VF Coeff | 0.05 |
| Entropy Coeff | 0.0 |
| PPO Clip | 0.2 |
| Grad Clip | 10.0 |

Table 11: MPE Simple Centroid Training Hyperparameters

## C.5. MPE Simple Tag + Noise

For the multi-agent experiments on the simple tag task, network architecture hyperparameters are displayed in Table 12 and training hyperparameters in Table 13.

| Hyperparameters | Value |
|---|---|
| Obs Shape | $N \times 8$ |
| Num Layers | 6 |
| Num Heads | 3 |
| Dim Hidden | 3 |
| Dim FF | 12 |
| Dropout Attn | 0.0 |
| Dropout FF | 0.1 |
| Bias Attn | False |
| Bias FF | True |
| Action MLP Layers | 3 |
| Action MLP Dim | 32 |
| Value MLP Layers | 3 |
| Value MLP Dim | 32 |
| Continuous Actions | False |

Table 12: MPE Simple Tag Network Hyperparameters

| Hyperparameters | Value |
|---|---|
| Algorithm | PPO |
| Num Seeds | 20 |
| Num Episodes | 500 |
| Training Batch | 8192 Agent Steps |
| Minibatch | 8192 Agent Steps |
| SGD Iter/Batch | 16 |
| Episode Length | 128 Env Steps |
| Learning Rate | $2.5e - 3$ |
| $\gamma$ | 0.99 |
| $\lambda$ | 0.95 |
| KL Coeff | 0.5 |
| KL Target | 0.01 |
| VF Coeff | 0.0005 |
| Entropy Coeff | 0.005 |
| PPO Clip | 0.2 |
| Grad Clip | 10.0 |

Table 13: MPE Simple Tag Training Hyperparameters

## C.6. BoxWorld

For the relational reasoning experiments in the BoxWorld environment, network architecture hyperparameters are displayed in Table 14 and training hyperparameters in Table 15. We also display images taken from the rendered environment in Figure 10. The learning agent controls the grey pixel. The agent's actions move it up, down, left, or right by one pixel on each step. Each round starts with a loose key, depicted by the lone blue pixel on the left side of Figure 10a. "Boxes" are indicated by pairs of colored pixels, with a lock on the right and the item in the box on the left. The agent must first pick up the loose key, which it can then use to open locks of the same color by moving onto them. The currently held key is displayed in a fixed position at the top left pixel, as shown in Figure 10b. The goal of the agent is

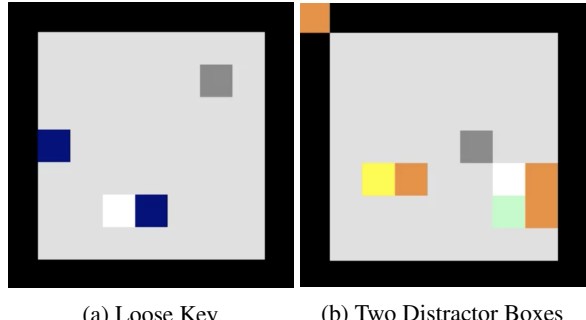

(a) Loose Key      (b) Two Distractor Boxes

Figure 10: (Left) The BoxWorld environment showing the agent, the initial loose key, and the box composed of a blue lock and the gem. (Right) A scenario containing two distractor paths, with the agent holding the orange key as visible in the top left.

to unlock the box with the white pixel referred to as the gem. In all training scenarios, we provide two distractor paths, shown by the additional boxes in Figure 10b. When it picks up the loose key, it earns +1 reward; when it unlocks the box with the gem, it earns +10 reward; if it opens a distractor box, it gets a penalty of -1.

| Hyperparameters | Value |
|---|---|
| Obs Shape | $N \times 5$ |
| Num Layers | 6 |
| Num Heads | 6 |
| Dim Hidden | 6 |
| Dim FF | 24 |
| Dropout Attn | 0.0 |
| Dropout FF | 0.1 |
| Bias Attn | False |
| Bias FF | True |
| Action MLP Layers | 3 |
| Action MLP Dim | 32 |
| Value MLP Layers | 3 |
| Value MLP Dim | 32 |
| Continuous Actions | False |

Table 14: BoxWorld Network Hyperparameters

| Hyperparameters | Value |
|---|---|
| Algorithm | PPO |
| Num Seeds | 5 |
| Num Episodes | 3000 |
| Training Batch | 14400 Agent Steps |
| Minibatch | 14400 Agent Steps |
| SGD Iter/Batch | 32 |
| Episode Length | 30 Env Steps |
| Learning Rate | $2.5e - 3$ |
| $\gamma$ | 0.99 |
| $\lambda$ | 0.95 |
| KL Coeff | 0.0 |
| VF Coeff | 0.01 |
| Entropy Coeff | 0.0 |
| PPO Clip | 0.2 |
| Grad Clip | 10.0 |

Table 15: BoxWorld Training Hyperparameters

## C.7. CIFAR Training Hyperparameters

All experiments trained vision transformers (ViT) from scratch on the CIFAR dataset using 5-fold cross-validation. As in the original paper, we note that vision transformers trained from scratch are less performant than similarly sized convolutional neural networks, as they lack spatial inductive biases, such as translation invariance (Dosovitskiy et al., 2021). Thus, performance reported in Table 1 is expected to be lower than a similarly sized ResNet trained from scratch on the same task, for example.

Training hyperparameters and network architecture details are outlined in Table 16. Each fold was trained for 300 epochs and, as in previous experiments, the network for each method was initialized with the same underlying transformer weights and seeds. Following the ViT paper, we apply gradient clipping at a global norm of $1.0$, as well as AdamW with a weight decay value of $0.1$. Finally, we use a learning rate scheduler with a linear warmup from $0.0$ to $0.001$ after 30 epochs (10% of training) and cosine annealing back down to $0.0$ by epoch 300.

| Hyperparameters | Value |
|---|---|
| Image Shape | $32 \times 32 \times 3$ |
| Patch Shape | $64 \times 4 \times 4 \times 3$ |
| Input Shape *(Flattened Patches)* | $64 \times 48$ |
| Num Layers | 6 |
| Num Heads | 8 |
| Dim Hidden | 512 |
| Dim FF | 64 |
| Dropout Attn | 0.1 |
| Dropout FF | 0.1 |
| Bias Attn | False |
| Bias FF | True |
| Training Epochs | 300 |
| Warmup Epochs | 30 |
| Learning Rate | 1.0e-3 |
| Batch Size | 512 |
| Weight Decay | 0.1 |
| Grad Clip | 1.0 |

Table 16: CIFAR network and training hyperparameters. Both experiments trained models from scratch and differed only by the number of output classes, 10 and 100 respectively.

