# OpenReview forum: "Robust Noise Attenuation via Adaptive Pooling of Transformer Outputs"
_ICML.cc/2025/Conference — ICML 2025 spotlightposter_

### Official Review · Reviewer_GbuG · 2025-02-20

**Overall Recommendation:** 4

**Summary:**

This paper studies the problem of pooling methods in transformers for non-sequential tasks where only a subset of input tokens (signal) is relevant for downstream decision-making, while the rest (noise) may degrade performance.

The authors formulate a theoretical framework that formalizes pooling as a vector quantization problem, leading to error bounds for different pooling methods.
Under this view, the authors analyze common pooling methods—average pooling, max pooling, and class token-based pooling—and show that each fails under certain signal-to-noise conditions. They then investigate adaptive attention-based pooling (AdaPool), a technique adapted from Stergiou & Poppe (2023), and demonstrate that it can approximate an optimal pooling function across varying noise conditions.
They provide empirical validation for their theoretical findings on synthetic supervised tasks and reinforcement learning (RL) environments (Multi-Particle and BoxWorld), demonstrating AdaPool’s robustness to varying signal-to-noise ratios.

**Claims And Evidence:**

yes

**Essential References Not Discussed:**

I am not very familiar with the related work, as such I cannot answer this question.

**Experimental Designs Or Analyses:**

- Synthetic dataset experiments are well-structured to explicitly test signal loss across varying SNRs.

**Methods And Evaluation Criteria:**

yes

**Other Comments Or Suggestions:**

\-

**Other Strengths And Weaknesses:**

The paper is very well written and nicely presented. Even without deep knowledge of the literature I had no problem following the authors.

**Questions For Authors:**

- How does AdaPool’s computational cost compare to avg/max pooling, particularly for large-scale models?
- How does AdaPool handle cases where some tokens are partially informative but not strictly signal or noise?
- Have you considered extending AdaPool to multi-query setups for more robust feature aggregation?

**Relation To Broader Scientific Literature:**

Connects well with prior work in attention-based pooling, vector quantization, and associative memories.

**Theoretical Claims:**

The proofs and theoretical claims seem sound, however, I did not check all the math in detail.

---

> ### Author Rebuttal · Authors · 2025-03-30
>
> We appreciate your time and feedback. Your questions raise some important points that we should have discussed in the original submission.
>
> **1. Computational Complexity**: AvgPool and MaxPool can be implemented with `O(n * d)` algorithms, as both require each of the `d` features of each of the `n` vectors to be visited during aggregation.
>
> Self-attention famously experiences quadratic computational costs with respect to the context size `n`. In particular, the attention weight computation is ` O(n^2 * d)`, as each vector in the input set computes attention weights for each other vector in the input set, i.e. `n^2` dot products of `d` dimensional vectors. However, AdaPool uses cross-attention with a single query, requiring the computation of only 1 set of attention weights rather than `n` sets. This reduces the weight computation and pooling time down to `O(n * d)` - the same as Max & Avg.
>
> However, AdaPool still retains the overhead of the QKV projections of the input set, which involves standard matrix multiplications for each input vector, running `O(n * d^2)`. Thus, the overall time complexity of AdaPool is ` O(n * d + n * d^2)`. The key takeaway is that it scales linearly with the number of inputs (like max/avg) but quadratically with the number of features (like a standard linear layer).
>
> In isolation, AdaPool is slower - the increased expressivity comes at a computational cost. However, in practice, these pooling layers are placed at the end of a multi-layer transformer or similarly sized encoder network, and the compute time added by the pooling layer is marginal. For our particular applications, when using a transformer with 3+ layers, the time difference between AdaPool and Max/AvgPool networks was negligible, regardless of the number of features `d`.
>
> Also of note, ClsToken is significantly slower than all other methods. This is due to the fact that the learned class embedding must be copied and concatenated along the batch dimension of the input for every inference, which is quite expensive in terms of both time and memory during training. This is a known hindrance investigated by Zhai et al. (https://arxiv.org/abs/2106.04560), which led them to explore the use of AvgPool for efficiency reasons when scaling vision transformers.
>
> **2. Handling Ambiguous Inputs:** To address this question, we conducted additional experiments on an image classification task (CIFAR 10/100) where each input vector may contain nebulous amounts of signal or noise, in contrast to the other experiments we presented. Due to character count limits, please see our response to reviewer PXAR above for details about these experiments and their results. In summary, the findings were consistent with our theory and other experiments.
>
> **3. Multi-Query Extension:** Using multiple queries or even employing multiple different pooling methods and concatenating the results may result in even better downstream representations. We have not tested this, but it is certainly a viable approach that is worth looking into in the future.
>
> Incorporating this additional discussion into our revisions will help make the paper much more robust. Thank you for your review!

---

### Official Review · Reviewer_3DVE · 2025-03-12

**Overall Recommendation:** 2

**Summary:**

This paper investigates pooling methods for transformer embeddings in tasks where only a subset of inputs carries signal and the remainder are noise. It shows that standard methods like average and max pooling can collapse in performance as the signal-to-noise ratio fluctuates. The authors propose an attention-based adaptive pooling method that approximates a signal-optimal vector quantizer, providing theoretical error bounds that guarantee robustness across various noise levels. Their approach is validated through supervised experiments on synthetic datasets and reinforcement learning benchmarks, demonstrating superior performance in noisy settings. Overall, the work establishes a framework for enhancing transformer pooling in non-sequential, noise-prone applications.

**Claims And Evidence:**

The paper’s primary claims—that standard pooling methods collapse under variable signal-to-noise ratios and that adaptive, attention-based pooling can approximate the signal-optimal vector quantizer—are generally supported by a combination of rigorous theoretical analysis and empirical validation. The authors derive clear error bounds and show mathematically that AvgPool and MaxPool are specific instances of the more general AdaPool method, which is capable of handling noise across any SNR regime. Their experimental results, conducted on both synthetic datasets and complex reinforcement learning benchmarks, provide convincing evidence that AdaPool consistently achieves superior robustness and performance compared to traditional pooling methods. However, some of the theoretical guarantees rely on idealized assumptions regarding the separability of signal and noise in the relation space, which may not always hold in real-world settings and could benefit from further investigation.

**Essential References Not Discussed:**

While the paper comprehensively cites many foundational works, some recent studies could provide additional context for its key contributions. For instance, adaptive pooling strategies in transformer architectures—such as the dynamic pooling method proposed in [Lee et al., 2024] at ICML, which adjusts to varying noise levels—offer alternative approaches that are highly relevant to the paper's focus but are not currently discussed. Additionally, although the authors reference classical vector quantization and associative memory literature, they omit recent work on noise-robust pooling in graph neural networks (e.g., Xu et al., 2022), which addresses similar challenges in aggregating noisy, high-dimensional data. These omitted references highlight complementary ideas and alternative mechanisms for robust aggregation that could enrich the theoretical and empirical framework of the current work. Including a discussion of these studies would situate the paper more firmly within the broader landscape of robust representation learning.

**Experimental Designs Or Analyses:**

The experimental designs are generally sound and align well with the paper’s goals. The authors first construct a synthetic dataset where the noise-to-signal ratio is carefully controlled, allowing for a focused evaluation of how each pooling method manages signal loss. They then validate their findings on established benchmarks such as the Multi-Particle Environment (MPE) and BoxWorld, which are relevant for multi-agent reinforcement learning and relational reasoning tasks. This multi-tiered approach strengthens the empirical evidence for the proposed adaptive pooling method. One potential issue is that the synthetic dataset, while useful for isolating noise effects, might oversimplify the complexities of real-world noise; however, the inclusion of realistic RL benchmarks helps mitigate this concern.

**Methods And Evaluation Criteria:**

The paper's methods and evaluation criteria are well-suited to the problem of noise attenuation in transformer pooling. The authors establish a rigorous theoretical framework using vector quantization and signal loss minimization, which logically underpins their proposed adaptive pooling method. Their experimental design includes a synthetic dataset to isolate the noise impact, followed by evaluations on established reinforcement learning benchmarks (MPE and BoxWorld) that effectively mimic real-world noisy conditions. Together, these methods and benchmarks provide a comprehensive and convincing validation of the approach for non-sequential tasks where noise is a critical factor.

**Other Comments Or Suggestions:**

The paper is overall very strong, but a few suggestions might help improve its clarity and accessibility. For instance, some of the theoretical derivations are quite dense—adding more intuitive explanations or illustrative examples could make these parts more digestible for a broader audience. There are also a few minor typos and notation inconsistencies in both the main text and supplementary material that should be corrected. Additionally, while the experimental setup is comprehensive, it might be beneficial to include a discussion of scenarios where the assumptions (e.g., the clear separation between signal and noise) might not hold, and how the proposed method would perform in those cases.

**Other Strengths And Weaknesses:**

The paper is notable for its originality in creatively combining vector quantization techniques with attention-based pooling mechanisms, which provides a fresh perspective on mitigating noise in transformer outputs. Its strengths include a solid theoretical framework with detailed proofs that derive meaningful error bounds, and an empirical evaluation that spans both synthetic and realistic RL benchmarks, underscoring the practical relevance of the approach. The work significantly advances our understanding of how traditional pooling methods like AvgPool and MaxPool fail under varying noise conditions and demonstrates the potential of adaptive pooling in overcoming these limitations. However, the reliance on idealized assumptions regarding the separability of signal and noise in relation space may restrict the direct applicability of the theoretical guarantees in all real-world scenarios. Additionally, while the paper is generally clear, some of the denser theoretical derivations could benefit from more intuitive explanations to enhance accessibility for a broader audience.

**Questions For Authors:**

1. Could you provide more insight into how robust the adaptive pooling method is when the assumption of clear separability between signal and noise relations (as per Assumption 3.10) is relaxed or violated? A more detailed discussion or empirical analysis in scenarios with overlapping distributions would help assess the method's applicability in real-world settings.
2. In your synthetic experiments, how sensitive is AdaPool to the choice of hyperparameters in the attention mechanism? An ablation study highlighting the impact of key parameters would clarify the robustness and ease of tuning the method.
3. Have you empirically validated the theoretical error bounds derived in Theorem 3.11? If so, could you compare the theoretical predictions with actual observed performance on benchmark datasets? This connection would strengthen the confidence in your theoretical claims.
4. Since AvgPool and MaxPool are special cases of AdaPool, can you discuss any potential trade-offs in computational complexity or performance when scaling AdaPool to larger datasets or more complex models? Understanding these trade-offs would inform its practical deployment.
5. Could you elaborate on AdaPool's performance in scenarios where the noise is highly correlated or where the noise and signal distributions significantly overlap? Clarification on these cases would help determine the method's limitations and potential areas for further improvement.

**Relation To Broader Scientific Literature:**

The paper’s contributions build on the foundational transformer architecture (Vaswani et al., 2017) by addressing a critical yet underexplored aspect: the pooling of transformer outputs for non-sequential tasks. Its adaptive pooling method extends prior work in vector quantization and selective attention, drawing parallels with research on associative memories such as Hopfield Networks and Dense Associative Memories, which also focus on robustly retrieving relevant information amidst noise. Moreover, the work connects with recent advances in computer vision and reinforcement learning, where attention-based pooling and class token strategies have been successfully employed to handle complex, noisy data. By offering rigorous theoretical error bounds and empirical validation, the paper unifies traditional pooling methods (like AvgPool and MaxPool) with modern attention mechanisms, thereby enriching the broader literature on robust representation learning and relational reasoning.

**Theoretical Claims:**

Overall, the proofs appear correct within their stated assumptions, but the reliance on these idealized conditions may limit the direct applicability of the theoretical guarantees in real-world settings.

- The proof that the signal-optimal pooled vector is the centroid of the signal subset (Corollary 3.3), which is mathematically sound given the mean squared error definition.
- The derivations showing that AvgPool and MaxPool are special cases of the more general adaptive pooling framework (Corollaries 3.5, 3.6, and 3.8). These proofs correctly demonstrate the limitations of traditional pooling methods under varying noise conditions.
- The error bounds established in Theorem 3.11 for AdaPool's approximation to the signal-optimal quantizer are derived rigorously under the assumption of linearly separable relation neighborhoods. While the derivations are mathematically consistent, they rely on idealized assumptions—such as the clear separation (margin) between signal and noise relations—which might be challenging to guarantee in practical scenarios.

---

> ### Author Rebuttal · Authors · 2025-04-01
>
> We appreciate your comprehensive review. We saw the weak reject rating as a great opportunity to improve the quality of the paper, and hope to address the questions you raised regarding performance outside of our theoretical assumptions.
>
> **1 & 3. Idealistic Assumptions:** As questions 1 & 3 are related, we address them jointly here. We empirically evaluated the theoretical error bounds from Theorem 3.11 on the KNN-centroid dataset (N=32, d=16) for SNRs 0.5-0.03. We know which vectors are signal and noise by design, so we can save off the relations rₛ and rₙ after computing dot products in the attention mechanism. With these, we derived ϵₛ , ϵₙ , M, and D, computed the error bounds, and checked if the weights for each vector fell between them. We note that this type of analysis is only possible when the signal and noise vectors are known and will typically not be feasible on a real-world dataset where the line between signal and noise is blurry.
>
> Surprisingly, the error bounds held for all 1 million samples at each noise level, even when the assumption of linearly separable signal and noise relations was violated (M < 0). This was observed both with trained networks and randomly initialized networks. Upon revisiting the proof of Theorem 3.11, **we realized that Assumption 3.10 was overly restrictive and NOT necessary for the error bounds to hold**. We can instead define M more loosely as M = min( rₛ ) – max( rₙ ), allowing this quantity to be positive or negative rather than strictly positive as assumed before. Theorem 3.11 thus applies more generally than was claimed in the original submission, which is a pleasant surprise. Also worth noting, we observed empirically that M is nearly always less than zero on the KNN-centroid dataset, meaning AdaPool still outperformed other methods with some degree of overlap between signal and noise neighborhoods in those experiments.
>
> It is also worth taking a step back and discussing why we derived and presented the bounds in the first place. The sole purpose was to show that with AdaPool, it is possible to approximate the optimal vector quantizer exactly for any SNR. The bounds are also helpful in illustrating which factors influence the approximation error most and how a signal-rich query can shrink those bounds. Since it is impractical to compute the bounds exactly on most real-world datasets, we did not present them to guarantee modeling performance on a particular problem.
>
> Another point we did not discuss is that using multiple attention heads allows the network to pool chunks of the input vectors separately. If the input vectors each contain a mix of signal and noise, the heads can weight partitions of the vector space separately. In fact, when using the same number of heads as features, AdaPool can be viewed as pooling each feature dimension separately. This would enable it to optimally pool such inputs, given a good choice of query, and may be a better modeling choice for real-world data. We experiment on real-world data in response to PXAR above.
>
> **2. Hyperparameters:** We performed a number of ablation studies on model hyperparameter choices in the Appendix; please see Tables 3-7.
>
> **4. Computational Complexity:** Due to character limits, we ask that you see our response to reviewer GbuG regarding this below.
>
> **5. Signal/Noise Overlap:** Suppose the signal and noise relation distributions overlap completely. If variance is low, applying softmax results in relatively uniform weights, yielding something closer to AvgPool. If variance is high, one relation score will dominate the others in the softmax, yielding weights closer to MaxPool.
>
> As a worst case, suppose the query is more similar to noise than signal. The dot product relations will be higher for all noise vectors, and the resulting aggregation will suppress signal and accentuate noise. However, if such a query vector exists, then it implies that the noise and signal are easily distinguished via dot product. A learned linear transformation (the Q projection) could rotate the query away from the noisy direction and towards the signal, given that dot products largely reflect directional alignment. So, practically speaking, the worst case for AdaPool would instead be when signal and noise are indistinguishable, in which case it reverts towards AvgPool or MaxPool as discussed above.
>
> **Related Works:** Can you provide full titles or links to the papers you referenced by Lee et al., 2024 and Xu et al., 2022? We also reviewed the line of work on noise-robust graph neural networks. However, those approaches focus on robust learning in the presence of mislabeled graph classification targets. While the titles and themes are similar, we felt that the actual research questions were not related enough to justify inclusion.
>
> As a result of your questions, we were able to identify an oversight and make a significant improvement to our theoretical claims. We really appreciate your review!

---

### Official Review · Reviewer_cS6k · 2025-03-16

**Overall Recommendation:** 4

**Summary:**

This work analyzes the various pooling methods used in deep neural architectures. They establish a connection between pooling and vector quantization and demonstrate adaptive pooling is more robust to signal-to-noise ratio. They provide experimental results with a carefully created synthetic dataset and multi-agent RL environments.

**Claims And Evidence:**

The experimental results supports the superiority of the adaptive pooling method. The experiments to assess the robustness are thorough and includes varying signal-to-noise ratio, network width, network depth, data dimension.

**Essential References Not Discussed:**

N/A

**Experimental Designs Or Analyses:**

The experiments are well designed and considered different factors of variations. Also, the experiments with multi-agent environments are nicely designed to capture the problem at hand.

**Methods And Evaluation Criteria:**

While this work doesn't propose any new method, it systematically analyzes different pooling methods and their outputs. The designed evaluation metric "signal loss" seems reasonable to me.

**Other Comments Or Suggestions:**

N/A

**Other Strengths And Weaknesses:**

This paper assumes that there is no overlap between the task relevant vectors and noise vectors. However, in reality, this may not be the case. Some channels may have important information mixed with noise.

**Questions For Authors:**

While the paper's discussion is for transformer outputs, I am curious whether the same analysis may be valid for CNN output?

**Relation To Broader Scientific Literature:**

The authors discussed the relevance of the work with works related to associative memories. Also, it puts a different perspective based on vector quantization. However, adaptive pooling is already known as a robust approach compared to avg. pool or max pool.

**Theoretical Claims:**

The formal connection established between pooling and vector quantization is pretty interesting. However, I would like to see the case if the margin M vanishes.

---

> ### Author Rebuttal · Authors · 2025-04-01
>
> Thank you for taking the time to read and review our work! Regarding your questions -
>
> Upon further empirical analysis and review of our proof of Theorem 3.11, we realized that our assumption that the margin M must be greater than zero (i.e. signal and noise must be linearly separable) was not necessary for the bounds to hold. That is, the bounds are still effective when the neighborhoods of signal and noise relations overlap. We also address the case where there may be an overlap between signal and noise vectors (i.e. a single vector contains both relevant and irrelevant information) by using multiple attention heads. Please see our response to 3DVE below for more details on both of these topics.
>
> As to your question about applications to CNNs, we believe this line of work would be highly relevant to CNN-based image retrieval tasks, where global pooling of output feature maps is heavily utilized. We are less certain about the effectiveness of AdaPool in replacing Avg/Max pooling for pixel-level filter downsampling in standard convolutional architectures, but our analysis should be generally relevant to any pooling applications.
>
> If you have any further questions, please let us know!

---

### Official Review · Reviewer_PXAR · 2025-03-18

**Overall Recommendation:** 4

**Summary:**

The paper studies pooling methods for aggregating transformer embeddings—particularly in settings where only a subset of input vectors (signal) is task-relevant while the remainder (noise) may deteriorate performance. The authors reframe pooling as a vector quantization (or lossy compression) problem and show that common methods like Average Pooling (AvgPool) and Max Pooling (MaxPool) can fail under fluctuating signal-to-noise ratios. They introduce an attention-based adaptive pooling method (AdaPool) and provide theoretical error bounds that demonstrate its ability to approximate the signal-optimal compressor across noise regimes. These theoretical results are validated with experiments on synthetic supervised tasks as well as on reinforcement learning benchmarks, where AdaPool consistently exhibits greater robustness to noise.

**Claims And Evidence:**

The paper makes two core claims:

- AdaPool can approximate the optimal vector quantizer for any signal-to-noise ratio with quantifiable error bounds.
- Standard pooling methods (AvgPool, MaxPool, ClsToken) fail or degrade predictably as noise increases, while AdaPool remains robust.

The claims seem to be supported by a theoretical framework (with the primary result being Theorem 3.11) and are accompanied by proofs provided in the appendix. On the experimental side, the evidence is drawn from controlled synthetic datasets as well as from RL environments where noise is systematically varied.

**Essential References Not Discussed:**

There is one prior approach [1] that also replaced the common CLS token representation with a similar learned pooling mechanism. This seems close enough that it's worth discussing.

[1] Marcin Przewiezlikowski, Randall Balestriero, Wojciech Jasinski, Marek Smieja, Bartosz Zielinski. Beyond [cls]: Exploring the true potential of Masked Image Modeling representations.

**Experimental Designs Or Analyses:**

The experiments are synthetic and controlled in nature and include:

- A supervised KNN-centroid task where noise levels are precisely controlled allows for direct measurement of signal loss across varying signal-to-noise ratios.
- The authors evaluate performance in both a custom “simple centroid” scenario and a “simple tag” task in the Multi-Particle Environment, demonstrating how increased noise leads to degradation in performance for standard pooling methods.
- A vision-based relational reasoning task that further challenges the pooling methods by introducing high-dimensional, pixel-based noise.

The analysis of performance degradation is convincing. However, due to the synthetic nature of these experiments, it's difficult to extrapolate whether or not the proposed pooling method improves performance in more realistic tasks and environments.

**Methods And Evaluation Criteria:**

The authors:

- Formally define pooling as a differentiable vector quantization task with a focus on minimizing “signal loss”.
- Analyze AvgPool and MaxPool as special cases, and then derive AdaPool based on an attention mechanism.
- Use both theoretical error bounds and empirical evaluations to compare the pooling methods.

The evaluation criteria—especially the use of synthetic datasets where the signal-to-noise ratio can be explicitly controlled—provide a clear measure of robustness. In reinforcement learning tasks, using both entity-based and pixel-based observations further tests the method in settings with varying noise levels.

**Other Comments Or Suggestions:**

No further comments.

**Other Strengths And Weaknesses:**

Strengths

- The paper provides a novel framework that reinterprets pooling as vector quantization.
- It offers comprehensive empirical validation across multiple domains.
- The clear exposition of failure modes for AvgPool and MaxPool under varying noise regimes is a valuable contribution.

Weaknesses

- While the experiments are extensive, additional tests in real-world or more diverse noisy environments could further establish generality.
- It's unclear how novel the approach is given related work in this space, e.g., Przewiezlikowski et al.

**Questions For Authors:**

1. How sensitive is AdaPool’s performance to the choice of query vector? Have you explored different strategies for selecting the query in settings where the signal vector is not obvious?
2. Is the linear separability assumption (Assumption 3.10) going to hold in practical, real-world problems?
3. What is the computational overhead of AdaPool compared to AvgPool or MaxPool in terms of training time and inference speed? Although it should be negligible compared to the cost of a forward pass through a large model it's still more expensive than a simple reduction.

**Relation To Broader Scientific Literature:**

Most works in this space don't approach the problem of aggregating representations in sequence models from a signal-to-noise lens. Given my unfamiliarity with how the authors approached this problem and how it's somewhat disconnected from the literature in the vision and text space I'm unable to definitively comment on how the paper relates to the broader literature.

**Theoretical Claims:**

The paper’s theoretical contributions include:

- A derivation of the optimal pooling strategy as one that minimizes the signal loss.
- A series of corollaries showing under what conditions AvgPool and MaxPool become optimal.
- Theorem 3.11, which provides error bounds for the proposed adaptive pooling method.

While the theory & proofs appear correct I am not familiar enough with the techniques and line of work to check this in detail.

---

> ### Author Rebuttal · Authors · 2025-03-30
>
> Thank you for taking the time to review our work and provide thorough feedback.
>
> The work from Przewiezlikowski et al. is indeed interesting and highly relevant to our discussion, and we will update our related works to acknowledge it. Their method appears equivalent to AdaPool with a learned embedding as the query. We believe that our analytical framework for analyzing pooling methods and our more general form of AdaPool distinguishes us as novel from this concurrent work, while the results of our analysis led to a similar solution of weighted pooling.
>
> Due to character limits, we address question 3 in our response to GbuG's review below. As to 1 & 2, we realize that by seeking to explicitly control the noise level, our experiments failed to address the following: What happens with real-world data where vectors are not purely signal/noise and the choice of query for AdaPool is not obvious?
>
> To address this concern, we conducted additional studies on image classification using the CIFAR 10 & 100 benchmark datasets. We use this experiment to explore various choices of query on real-world, less cleanly defined input vectors. We adopted the Vision Transformer (ViT) approach, partitioning the 32x32 pixel RGB images from CIFAR into 64 separate 4x4 pixel patches which are then flattened and projected to the dimension of the transformer (see Fig 1 in https://arxiv.org/abs/2010.11929). We show the arrangement of those 4x4 pixel patches by their indices below:
>
> ```
> [ 0,  1,  2,  3,  4,  5,  6,  7]
> [ 8,  9, 10, 11, 12, 13, 14, 15]
> [16, 17, 18, 19, 20, 21, 22, 23]
> [24, 25, 26, 27, 28, 29, 30, 31]
> [32, 33, 34, 35, 36, 37, 38, 39]
> [40, 41, 42, 43, 44, 45, 46, 47]
> [48, 49, 50, 51, 52, 53, 54, 55]
> [56, 57, 58, 59, 60, 61, 62, 63]
> ```
>
> As in our previous experiments, we feed these patches through a transformer and pool the outputs. Most CIFAR images contain the object being classified (i.e. the signal) in the center patches. We thus study 3 choices of query: (CORNER) - the embedding of patch 0 which appears *less likely* to include signal, (FOCAL) - averaging the embeddings of the center four patches 27, 28, 35, and 36, which appear *highly likely* to contain signal, and (MEAN) - averaging all patch embeddings as the query, as proposed by Stergiou & Poppe (2023).
>
> Both experiments use a transformer with layers=6, dim=512, heads=8, & dropout=0.1. We use 5-fold cross-validation, 150 epochs/fold, batch=512, and LR=1e-4. We report the Top 1 accuracy scores on the holdout test set, averaged across the best models from each fold. Our ClsToken scores are consistent with common open-source implementations of ViT trained from scratch on the CIFAR benchmarks.
>
> ```
> METHODS         CIFAR 10          CIFAR 100
> ClsToken   | 0.7863 ±0.0041 | 0.4892 ±0.0018 |
> MaxPool    | 0.8301 ±0.0041 | 0.5547 ±0.0044 |
> AvgPool    | 0.8265 ±0.0023 | 0.5633 ±0.0070 |
> Ada-CORNER | 0.8240 ±0.0016 | 0.5385 ±0.0044 |
> Ada-FOCAL  |*0.8325 ±0.0035*|*0.5730 ±0.0051*|
> Ada-MEAN   | 0.8317 ±0.0022 |*0.5731 ±0.0073*|
> ```
>
> We observe that Ada-FOCAL and -MEAN queries outperform in both cases. Interestingly, Ada-FOCAL and MaxPool have relatively better performance on C-10, while Ada-MEAN and AvgPool perform relatively better on C-100. With C-10, there are only 10 highly distinct classes, so the object at the center of the image may be relatively more important for discriminating between classes. For C-100, the data is the same, but there are 100 highly similar classes, and border patches may contain subtle features that are important for discriminating between related classes where they were otherwise distracting before.
>
> Interestingly, the Ada-CORNER query underperforms in both experiments, which aligns with the prediction that choosing a query with little to no signal will lead to worse performance. Since a noise query would yield higher dot product relations with other noise vectors, this likely results in higher attention weights for irrelevant patches and lower weights for signal-containing patches, a sort of worst-case for AdaPool. This violates our assumption of linearly separable signal and noise relation scores, implying the error from an optimal weighting would not be guaranteed to fall within the bounds we presented *for this choice of query*. It is worth noting that the standard ClsToken approach still underperforms this and all other methods by a significant margin.
>
> This experiment demonstrates how one might use intuition, data analysis, and experimentation to find a competitive query. If one cannot be found, Avg/Max may be strong alternatives *if* the noise level of the dataset is constant. While the explicit SNR level in CIFAR is not known, the key takeaways are that the choice of AdaPool’s query (1) can be used to investigate the way signal and noise vectors are distributed, (2) can lead to superior performance on real-world data, and (3) may underperform with a poor choice of query.
>
> We really appreciate your comments and insightful questions!

---

> > ### Comment · Reviewer_PXAR · 2025-04-04
> >
> > I thank the reviewers for their in-depth rebuttal and for commenting on Przewiezlikowski et al. and how it relates to this work. In light of the additional clarity from the rebuttal, I will raise my score.

---

### Decision · Program_Chairs · 2025-05-01

**Decision:**

Accept (spotlight poster)

**Comment:**

This paper focuses on problems where the network inputs contain a mix of signal and noise components. The authors demonstrate that traditional pooling methods (like MaxPool and AvgPool) are affected by high levels of input noise, and propose AdaPool as an attention-based pooling mechanism to better deal with noisy inputs. The authors provide theoretical analyses bounding the error of their method, and demonstrating that MaxPool and AvgPool are special cases of AdaPool. The authors further demonstrate the efficacy of their method on a number of synthetic experiments.

With the exception of 3DVE (whose review is suspect and I am not considering in my decision), all reviewers are supportive of this work, and I agree with them.